# Variance predicts salience in central sensory processing

**Ann M Hermundstad[1,2]\*, John J Briguglio[1], Mary M Conte[3], Jonathan D Victor[3†], Vijay Balasubramanian[1,2,5†], Gašper Tkačik[4†]**

[1]Department of Physics and Astronomy, University of Pennsylvania, Philadelphia, United States; [2]Laboratoire de Physique Théorique, École Normale Supérieure, Paris, France; [3]Brain and Mind Research Institute, Weill Cornell Medical College, New York, United States; [4]Institute of Science and Technology Austria, Klosterneuburg, Austria; [5]Initiative for the Theoretical Sciences, City University of New York Graduate Center, New York, United States

**Abstract** Information processing in the sensory periphery is shaped by natural stimulus statistics. In the periphery, a transmission bottleneck constrains performance; thus efficient coding implies that natural signal components with a predictably wider range should be compressed. In a different regime—when sampling limitations constrain performance—efficient coding implies that *more* resources should be allocated to informative features that are more variable. We propose that this regime is relevant for sensory cortex when it extracts complex features from limited numbers of sensory samples. To test this prediction, we use central visual processing as a model: we show that visual sensitivity for local multi-point spatial correlations, described by dozens of independently-measured parameters, can be quantitatively predicted from the structure of natural images. This suggests that efficient coding applies centrally, where it extends to higher-order sensory features and operates in a regime in which sensitivity increases with feature variability.

**\*For correspondence:**
annherm@physics.upenn.edu

†These authors contributed equally to this work

**Competing interests:** The authors declare that no competing interests exist.

**Reviewing editor**: Timothy Behrens, Oxford University, United Kingdom

## Introduction

Sensory receptor neurons encode signals from the environment, which are then transformed by successive neural layers to support diverse and computationally complex cognitive tasks. A normative understanding of these computations begins in the periphery, where the efficient coding principle—the notion that a sensory system is tuned to the statistics of its natural inputs—has been shown to be a powerful organizing framework (*Barlow, 2001*; *Simoncelli, 2002*). Perhaps the best-known example is that of redundancy removal via predictive coding and spatiotemporal decorrelation. In insects, this is carried out by neural processing (*Laughlin, 1981*; *van Hateren, 1992b*); in vertebrates, fixational eye movements—which precede the first step of neural processing (*Srinivasan et al., 1982*; *Atick and Redlich, 1990*; *Atick et al., 1992*)—play a major role (*Kuang et al., 2012*). This approach was later extended to describe population coding, retinal mosaic structure (*Barlow, 2001*; *Karklin and Simoncelli, 2001*; *Borghuis et al., 2008*; *Balasubramanian and Sterling, 2009*; *Liu et al., 2009*; *Garrigan et al., 2010*; *Ratliff et al., 2010*; *Kuang et al., 2012*), adaptation of neural responses (*Brenner et al., 2000*; *Fairhall et al., 2001*; *Schwartz and Simoncelli, 2001*), and early auditory processing (*Smith and Lewicki, 2006*). Taken together, normative theories based on efficient coding have been successful in explaining aspects of processing in the sensory periphery that are tuned to simple statistical features of the natural world.

Can we extend such theories beyond the sensory periphery to describe cortical sensitivity to complex sensory features? Normative theories have been successful in predicting the response properties of single cells, including receptive fields in V1 (*Olshausen and Field, 1996*; *Bell and Sejnowski, 1997*;

**eLife digest** Our senses are constantly bombarded by sights and sounds, but the capacity of the brain to process all these inputs is finite. The stimuli that contain the most useful information must therefore be prioritized for processing by the brain to ensure that we build up as complete a picture as possible of the world around us. However, the strategy that the brain uses to select certain stimuli—or certain features of stimuli—for processing at the expense of others is unclear.

Hermundstad et al. have now provided new insights into this process by analyzing how humans respond to artificial stimuli that contain controllable mixtures of features that found in natural stimuli. To do this, Hermundstad et al. selected photographs of the natural world, and measured the brightness of individual pixels. After adjusting images in a way that mimics the human retina, the brightest 50% of the pixels in each photograph were colored white and the remaining 50% were colored black.

Hermundstad et al. then used statistical techniques to calculate the degree to which the color of pixels could be used to predict the color of their neighbors. In this way, it was possible to calculate the amount of variation throughout the images, and then make computer-generated images in which pixel colorings were more or less predictable than in the natural images.

Volunteers then performed a task in which they had to locate a computer-generated pattern against a background of random noise. The volunteers were able to locate this target most easily when it contained the same kinds of patterns and features that were meaningful about natural images.

While this shows that the brain is adapted to prioritize features that are more informative about the natural world, understanding exactly how the brain implements this strategy remains a challenge.

*van Hateren and Ruderman, 1998*; *van Hateren and van der Schaaf, 1998*; *Hyvarinen and Hoyer, 2000*; *Vinje and Gallant, 2000*; *Karklin and Lewicki, 2009*) and spectro-temporal receptive fields in primary auditory cortex (*Carlson and DeWeese, 2002, 2012*), as well as distributions of tuning curves across individual cells in a population (*Lewicki, 2002*; *Ganguli and Simoncelli, 2011*). Some complex features, however, might not be represented by the tuning properties of individual cells in any direct way, but rather emerge from the collective behavior of many cells. Instead of trying to predict individual cell properties, we therefore focus on the sensitivity of the complete neural population. Is there an organizing principle that determines how resources within the population are allocated to representing such complex features?

When the presence of complex features is predictable (i.e., can be accurately guessed from simpler features along with priors about the environment), mechanisms are best devoted elsewhere (See Discussion, *van Hateren, 1992a*). In contrast, sensory features that are highly variable and not predictable from simpler ones can serve to determine their causes (e.g., to distinguish among materials or objects), a first step in guiding decisions. We will show that these ideas predict a specific organizing principle for aggregate sensitivities arising in cortex: the perceptual salience of complex sensory signals increases with the variability, or unpredictability, of the corresponding signals over the ensemble of natural stimuli.

To test this hypothesis, we focus on early stages of central visual processing. Here, early visual cortex (V1 and V2) is charged with extracting edges, shapes, and other complex correlations of light between multiple points in space (*Morrone and Burr, 1988*; *Oppenheim and Lim, 1981*; *von der Heydt et al., 1984*). We compare the spatial variation of local patterns of light across natural images with human sensitivity to manipulations of the same patterns in synthetic images. This allows us to determine how sensitivity is distributed across many different features, rather than simply determining the most salient ones. (We will say that a feature is more *salient* if it is more easily discriminated from white noise.) To this end, we parametrize the space of local multi-point correlations in images in terms of a complete set of coordinates, and we measure the probability distribution of coordinate values sampled over a large ensemble of natural scenes. We then use a psychophysical discrimination task to measure human sensitivity to the same correlations in synthetic images, where the correlations can be isolated and manipulated in a mathematically rigorous fashion by varying the corresponding coordinates (*Chubb et al., 2004*; *Victor et al., 2005*; *Victor and Conte, 2012*; *Victor et al., 2013*). Comparing the measurements, we show that human sensitivity to these multi-point elements of visual

form is tuned to their variation in the natural world. Our result supports a broad hypothesis: cortex invests preferentially in mechanisms that encode unpredictable sensory features that are more variable, and thus more informative about the world. Namely, *variance is salience*.

## Results

As we recently showed, some informative local correlations of natural scenes are captured by the configurations of luminances seen through a 'glider', that is, a window defined by a 2 × 2 square arrangement of pixels (*Tkačik et al., 2010*). We use this observation first as a framework for analyzing the local statistical structure of natural scenes, then to characterize psychophysical sensitivities via a set of synthetic visual texture stimuli, and finally to compare the two.

### Analyzing local image statistics in natural scenes

The analysis of natural scenes is schematized in *Figure 1*. We collect an ensemble of image patches from the calibrated Penn natural image database (PIDB) (*Tkačik et al., 2011*). We preprocess the image patches as shown in *Figure 1A*. This involves first averaging pixel luminances over a square region of $N \times N$ pixels, which converts an image of size $L_1 \times L_2$ pixels into an image of reduced size $L_1/N \times L_2/N$ pixels. Images are then divided into $R \times R$ square patches of these downsampled pixels and whitened (see 'Materials and methods', *Image preprocessing*, for further details). Since the preprocessing depends on a choice of two parameters, the block-average factor $N$ and patch size $R$, we report results for multiple image analyses performed using the identical preprocessing pipeline but for various choices of $N$ and $R$. After preprocessing, we binarize each patch to have equal numbers of black and white pixels (black = −1, white = +1). We characterize each patch by the histogram of 16 binary colorings ($2^{2\times2}$) seen through a square 2 × 2 pixel glider (*Figure 1B*). Translation invariance imposes constraints on this histogram, reducing the number of degrees of freedom to 10 (*Victor and Conte, 2012*). These degrees of freedom can be mapped to a set of image statistic coordinates that separates correlations based on their order: (*i*) one first-order coordinate, $\gamma$, describes overall luminance, (*ii*) four second-order coordinates, $\{\beta_|, \beta_-, \beta_/, \beta_\backslash\}$, describe two-point correlations between pixels arranged vertically, horizontally, or diagonally, (*iii*) four third-order coordinates, $\{\theta_\llcorner, \theta_\lrcorner, \theta_\urcorner, \theta_\ulcorner\}$, describe three-point correlations between pixels arranged into ∟-shapes of different orientations, and (*iv*) one fourth-order coordinate, $\alpha$, describes the single four-point correlation between all four pixels in the glider (*Figure 1C*). The binarization step of the preprocessing pipeline forces $\gamma$ to zero, leaving nine coordinates. Each image patch is thus characterized by a vector of coordinate values $\{\beta_|, \beta_-, \beta_/, \beta_\backslash, \theta_\lrcorner, \theta_\urcorner, \theta_\ulcorner, \theta_\llcorner, \alpha\}$, that is, a point within the multidimensional space of image statistics. Accumulating these points across patches yields a multidimensional probability distribution that characterizes the local correlations in natural scenes (schematized in *Figure 1D*). A total of 724 images (up to 249780 patches, depending on the choice of $N$ and $R$), was used to construct this distribution.

To summarize this distribution, we compute the degree of variation (standard deviation) along each coordinate axis (*Figure 1E*). As is shown, the degree of variation along different coordinate axes exhibits a characteristic rank-ordering, given by $\{\beta_|, \beta_-\} > \{\beta_/, \beta_\backslash\} > \alpha > \{\theta_\llcorner, \theta_\lrcorner, \theta_\urcorner, \theta_\ulcorner\}$; that is, the most variable correlations are pairwise correlations in the cardinal directions, followed by pairwise correlations in the oblique directions, followed by fourth-order correlations. Interestingly, third-order correlations are the least variable across image patches. An analogous analysis performed on white noise yields a flat distribution with considerably smaller standard deviation values (See 'Materials and methods', *Analysis variants for Penn Natural Image Database*, and *Figure 1—figure supplement 3* for comparison), and performing the analysis on a colored Gaussian noise (e.g. $1/f^k$ spectrum) would also yield a flat distribution because of the whitening stage in the image preprocessing pipeline. These (and subsequent) findings are preserved across different choices of image analysis parameters (shown in *Figure 1E* for block-average factors $N = 2, 4$ and patch sizes $R = 32, 48, 64$; see 'Materials and methods', *Analysis variants for Penn Natural Image Database*, and *Figure 3—figure supplement 5A* for a larger set of parameters) and also across other collections of natural images (see 'Materials and methods', *Comparison with van Hateren Database*, and *Figure 3—figure supplement 5B* for a parallel analysis of the van Hateren image dataset (*van Hateren and van der Schaaf, 1998*), which gives similar results).

### Characterizing visual sensitivity to local image statistics

To characterize perceptual sensitivity to different statistics, we isolated them in synthetic visual images and used a figure/ground segmentation task (*Figure 2B*). We used a four-alternative forced-choice

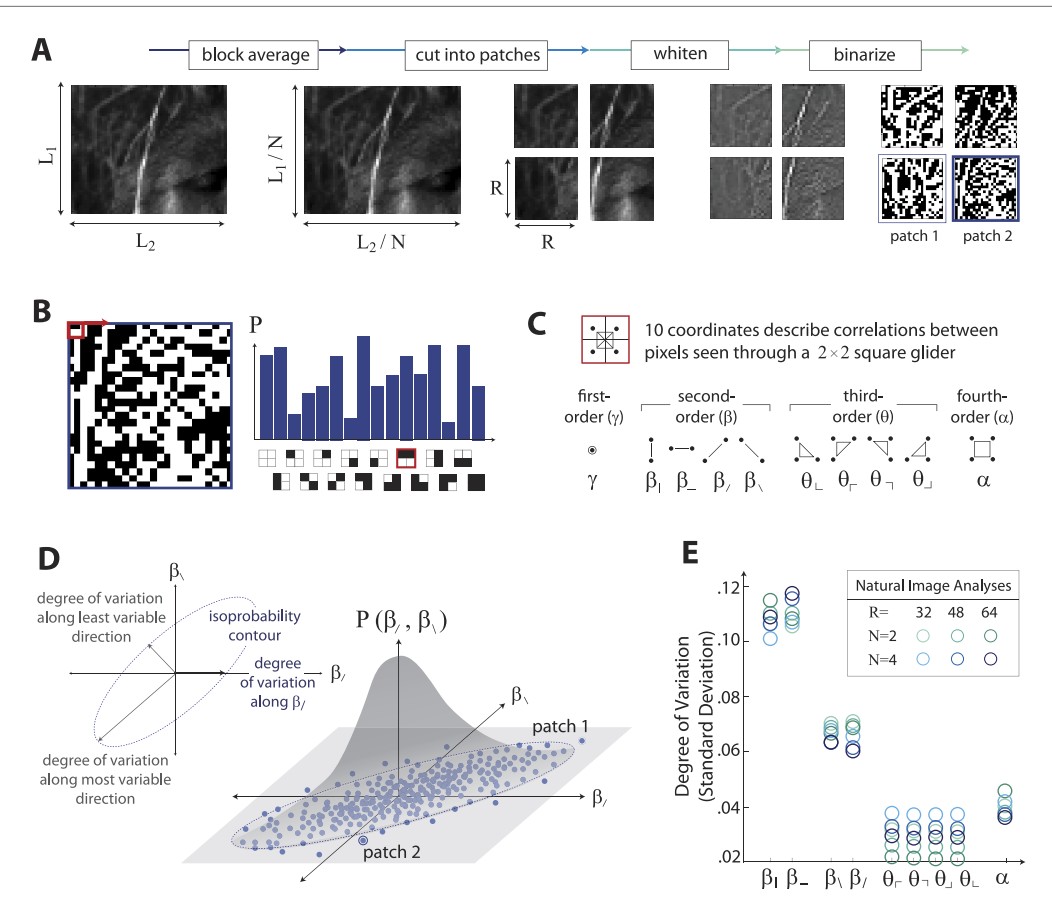

**Figure 1**. Extracting image statistics from natural scenes. (**A**) We first block-average each image over $N \times N$ pixel squares, then divide it into patches of size $R \times R$ pixels, then whiten the ensemble of patches by removing the average pairwise structure, and finally binarize each patch about its median intensity value (see 'Materials and methods', *Image preprocessing*). (**B**) From each binary patch, we measure the occurrence probability of the 16 possible colorings as seen through a two-by-two pixel glider (red). Translation invariance imposes constraints between the probabilities that reduce the number of degrees of freedom to 10. (**C**) A convenient coordinate basis for these 10° of freedom can be described in terms of correlations between pixels as seen through the glider. These consist of one first-order coordinate ($\gamma$), four second-order coordinates ($\beta_|,\beta_-,\beta_/,\beta_\backslash$), four third-order coordinates ($\theta_\llcorner,\theta_\ulcorner,\theta_\urcorner,\theta_\lrcorner$), and one fourth-order coordinate ($\alpha$). Since the images are binary, with black = −1 and white = +1, these correlations are sums and differences of the 16 probabilities that form the histogram in panel B (***Victor and Conte, 2012***). (**D**) Each patch is assigned a vector of coordinate values that describes the histogram shown in (**B**). This coordinate vector defines a specific location in the multidimensional space of image statistics. The ensemble of patches is then described by the probability distribution of coordinate values. We compute the degree of variation (standard deviation) along different directions within this distribution (inset). (**E**) Along single coordinate axes, we find that the degree of variation is rank-ordered as $\{\beta_|,\beta_-\} > \{\beta_/,\beta_\backslash\} > \alpha > \{\theta_\llcorner,\theta_\ulcorner,\theta_\urcorner,\theta_\lrcorner\}$, shown separately for different choices of the block-average factor $N$ and patch size $R$ used during image preprocessing.

The following figure supplements are available for figure 1:

**Figure supplement 1**. Two-component decomposition of natural image distribution.

**Figure supplement 2**. Filtering via defocus or motion blur reassigns sharp image patches to the 'blurry' component.

**Figure supplement 3**. Image statistics along single coordinate axes for white-noise patches.

task in which stimuli consisted of a textured target and a binary noise background (or vice-versa). Each stimulus was presented for 120ms and was followed by a noise mask. Subjects were then asked to identify the spatial location (top, bottom, left, or right) of the target. Experiments were carried out for synthetic stimuli in which the target or background was defined by first varying image statistic coordinates independently (*Figure 2A* shows examples of gamuts from which stimuli are built). Along each coordinate axis, threshold (1/sensitivity) was defined as the coordinate value required to support a criterion level of performance (*Figure 2C*, inset). We then performed further experiments in which the target or background was defined by simultaneously varying pairs of coordinates. For measurements involving each coordinate pair (to which we will refer as a 'coordinate plane'), we traced out an isodiscrimination contour (*Figure 2C*) that describes the threshold values not only along the cardinal directions, but also along oblique directions. Measurements were collected for four individual subjects in each of 11 distinct coordinate planes (representing all distinct coordinate pairs up to 4-fold rotational symmetry; see 'Materials and methods', *Psychophysical methods*, for further details). Each subject performed 4320 judgements per plane, for a total of 47,520 trials per subject.

*Figure 2D* shows perceptual sensitivities measured along each coordinate axis. For each of four subjects, a similar pattern emerges for sensitivities as was observed for variation in natural image statistics: sensitivities are rank-ordered as $\{\beta_|, \beta_-\} > \{\beta_/, \beta_\backslash\} > \alpha > \{\theta_\ulcorner, \theta_\urcorner, \theta_\lrcorner, \theta_\llcorner\}$.

Note that the difference between the sensitivities in the horizontal and vertical directions ($\beta_-$ and $\beta_|$) vs the diagonal directions ($\beta_\backslash$ and $\beta_/$) is not simply an 'oblique effect', that is, a greater sensitivity to cardinally- vs obliquely-oriented contours (*Campbell et al., 1966*). Horizontal and vertical pairwise correlations differ from the diagonal pairwise correlations in more than just orientation: pixels involved in horizontal and vertical pairwise correlations share an edge, while pixels involved in diagonal pairwise correlations only share a corner. Correspondingly, the difference in sensitivities for horizontal and vertical correlations vs diagonal correlations is approximately 50%, which is much larger than the size of the classical oblique effect (10–20%) (*Campbell et al., 1966*).

## Natural scenes predict human sensitivity along single coordinates

*Figures 1E and 2D* show a rank-order correspondence between natural image statistics and perceptual sensitivities. This qualitative comparison can be converted to a quantitative one (*Figure 3A*), as a single scaling parameter aligns the standard deviation of natural image statistics with the corresponding perceptual sensitivities. In this procedure, each of the six image analyses is scaled by a single multiplicative factor that minimizes the squared error between the set of standard deviations and the set of subject-averaged sensitivities (see 'Materials and methods', *Image preprocessing*, and *Figure 3—figure supplement 1* for additional details regarding scaling). The agreement is very good, with the mismatch between image analyses and human psychophysics comparable to the variability from one image analysis to another, or from one human subject to another.

We quantify the correspondence between image analyses and psychophysical analyses by computing the scalar product between the normalized vector of standard deviations (extracted separately from each image analysis) and the normalized vector of subject-averaged sensitivities (extracted from the set of psychophysical analyses). A value of 1 indicates perfect correspondence, and 0 indicates no correspondence. This value ranges from 0.987 to 0.999 across image analyses and is consistently larger than the value measured under the null hypothesis that the apparent correspondence between statistics and sensitivities is chance ($p \leq .0003$ for each image analysis; see *Tables 1–2* and 'Materials and methods', *Permutation tests*, for details regarding statistical tests).

These findings support our hypothesis that human perceptual sensitivity measured along single coordinate axes (assessed using synthetic binary textures) is predicted by the degree of variation along the same coordinate axes in natural scenes.

## Natural scenes predict human sensitivity to joint variations of all pairs of coordinates

The correspondence shown in *Figure 3A* considers each image statistic coordinate in isolation. However, it is known that image statistics covary substantially in natural images (as diagrammed in *Figure 1D*) and also that they interact perceptually (as diagrammed in *Figure 2C*). When pairs of natural image statistics covary, thus sampling oblique directions not aligned with the coordinate axes in the space of image statistics, our hypothesis predicts that human perceptual sensitivity is matched to both the degree and the direction of that covariation (we are referring here to the orientation of a

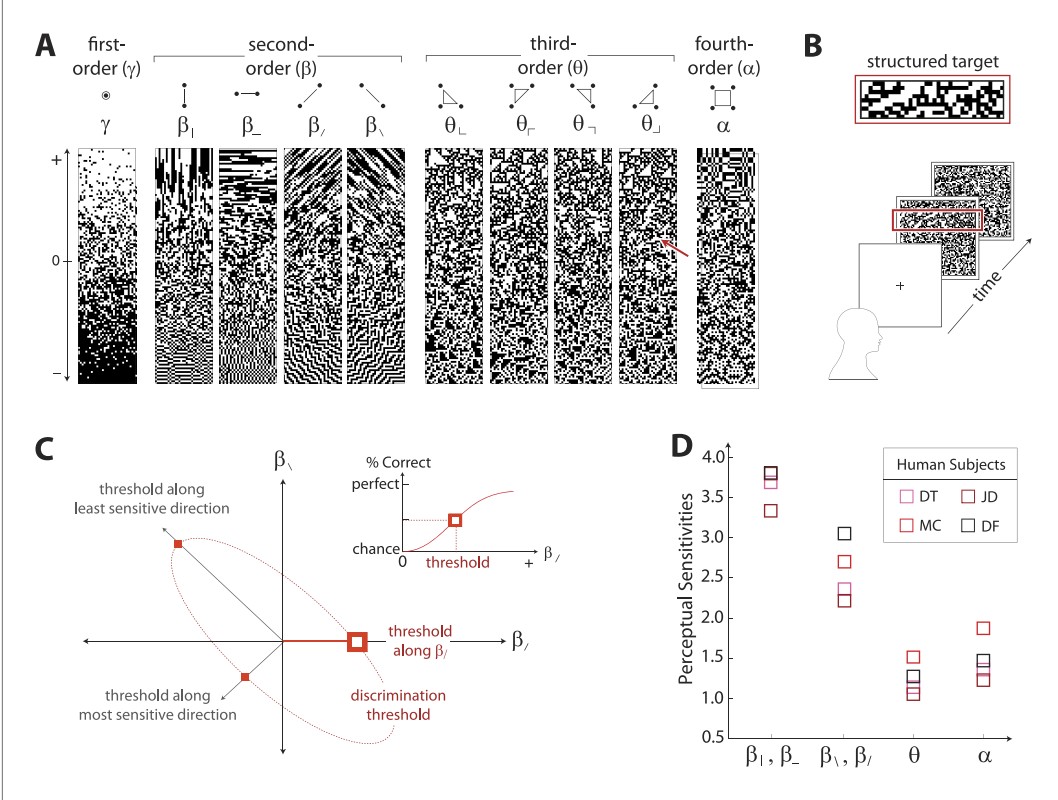

**Figure 2**. Measuring human sensitivity to image statistics. (**A**) Synthetic binary images can be created that contain specified values of individual image statistic coordinates (as shown here) or specified values of pairs of coordinates (***Victor and Conte, 2012***). (**B**) To measure human sensitivity to image statistics, we generate synthetic textures with prescribed coordinate values but no additional statistical structure, and we use these synthetic textures in a figure/ground segmentation task (See ***Victor and Conte, 2012*** and 'Materials and methods', *Psychophysical methods*). (**C**) For measurements along coordinate axes, test stimuli are built out of homogeneous samples drawn from the gamuts shown in **A** (e.g. the target shown in B was generated from the portion of the gamut indicated by the red arrow in **A**; See 'Materials and methods', *Psychophysical methods*, and ***Victor et al., 2005***; ***Victor and Conte, 2012***; ***Victor et al., 2013***). We assess the discriminability of these stimuli from white noise by measuring the threshold value of a coordinate required to achieve performance halfway between chance and perfect (inset). A similar approach is used to measure sensitivity in oblique directions; here, two coordinate values are specified to create the test stimuli. The threshold values along the axes and in oblique directions define an isodiscrimination contour (red dashed ellipse, main panel) in pairwise coordinate planes. (**D**) Along individual coordinate axes, we find that sensitivities (1/thresholds) are rank-ordered as $\left\{\beta_|, \beta_-\right\} > \left\{\beta_/, \beta_\backslash\right\} > \alpha > \left\{\theta_\Gamma, \theta_\neg, \theta_\lrcorner, \theta_\llcorner\right\}$, shown separately for four individual subjects. A single set of perceptual sensitivities is shown for $\left(\beta_|, \beta_-\right), \left(\beta_/, \beta_\backslash\right)$, and $\left(\theta_\llcorner, \theta_\Gamma, \theta_\neg, \theta_\lrcorner\right)$, since human subjects are equally sensitive to rotationally-equivalent pairs of second-order coordinates and to all third-order coordinates (***Victor et al., 2013***).

distribution in the coordinate plane of a pair of image statistics, and not to an orientation in physical space). To test this idea, we proceeded as follows.

First, we fit the distribution of image statistics with a multidimensional Gaussian. When projected into pairwise coordinate planes, the isoprobability contours of this Gaussian capture the in-plane shape and orientation of the covariation of the distribution. Along single coordinate axes, the variation in natural image statistics predicts human perceptual sensitivities, as we have shown (***Figure 3A***). More generally, we would predict that sensitivity should be be high along directions in which the distribution of natural image statistics has high standard deviation, because in those directions, the position of a sample cannot be guessed. Within coordinate planes, the quantitative statement of this idea is that the inverse covariance matrix, or precision matrix, predicts perceptual isodiscrimination contours. Sensitivity is expected to be low (and therefore threshold high) along

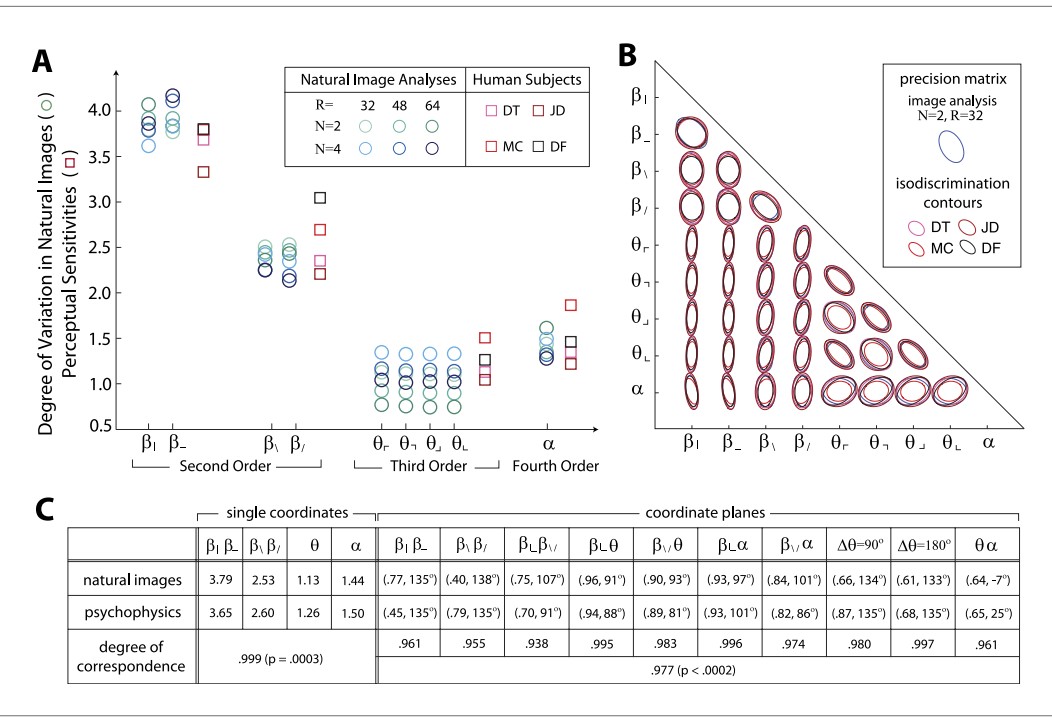

**Figure 3**. Variation in natural images predicts human perceptual sensitivity. (**A**) Scaled degree of variation (standard deviation) in natural image statistics along second- ($\beta$), third- ($\theta$), and fourth-order ($\alpha$) coordinate axes (blue circular markers) are shown in comparison to human perceptual sensitivities measured along the same coordinate axes (red square markers). Degree of variation in natural image statistics is separately shown for different choices of the block-average factor (*N*) and patch size (*R*) used during image preprocessing. Perceptual sensitivities are separately shown for four individual subjects. As in **Figure 2C,A** single set of perceptual sensitivities is shown for $(\beta_|, \beta_-)$, $(\beta_/, \beta_\backslash)$, and $(\theta_\llcorner, \theta_\ulcorner, \theta_\urcorner, \theta_\lrcorner)$. (**B**) For each pair of coordinates, we compare the precision matrix (blue ellipses) extracted from natural scenes (using *N* = 2, *R* = 32) to human perceptual isodiscrimination contours (red ellipses). Coordinate planes are organized into a grid. The set of ellipses in each pairwise plane is scaled to maximally fill each portion of the grid; agreement between the variation along single coordinate axes and the corresponding human sensitivities (shown in **A**) guarantees that no information is lost by scaling. Across all 36 coordinate planes, there is a correspondence in the shape, size, and orientation of precision matrix contours and perceptual isodiscrimination contours. (**C**) Quantitative comparison of a single image analysis (*N* = 2, *R* = 32) with the subject-averaged psychophysical data. For single coordinates depicted in A, we report the standard deviation in natural image statistics (upper row) and perceptual sensitivities (middle row). For sets of coordinate planes depicted in (**B**), we report the (average eccentricity, angular tilt) of precision matrix contours from natural scenes (upper row) and isodiscrimination contours from psychophysical measurements (middle row). The degree of correspondence between predictions derived from natural image data and the psychophysical measurements can be conveniently summarized as a scalar product (see text), where 1 indicates a perfect match. In all cases, the correspondence is very high (0.938–0.999) and is highly statistically significant ($p \le 0.0003$ for both single coordinates and pairwise coordinate planes; see 'Materials and methods', *Permutation tests*, for details).

The following figure supplements are available for figure 3:

**Figure supplement 1**. Scaling of natural image analyses.

**Figure supplement 2**. Covariation in natural image statistics predicts human isodiscrimination contours.

**Figure supplement 3**. Principal axes of variation in natural images predict principal axes of perceptual sensitivity.

**Figure supplement 4**. Mapping ellipse shapes to the quarter unit sphere.

**Figure supplement 5**. Single coordinate axes: variation in natural images predicts human perceptual sensitivities.

*Figure 3. Continued on next page*

*Figure 3. Continued*

**Figure supplement 6**. Pairwise coordinate planes in Penn Natural Image Database: covariation in natural images predicts human isodiscrimination contours.

**Figure supplement 7**. Pairwise coordinate planes in van Hateren Image Database: covariation in natural images predicts human isodiscrimination contours.

**Figure supplement 8**. Principal axes of variation across natural images predict principal axes of human perceptual sensitivity in the full coordinate space.

**Figure supplement 9**. Asymmetries in natural image statistics.

directions in which the precision matrix has a high value and the position of a sample can be guessed a priori.

Results in each coordinate plane are shown in *Figure 3B*. Across all subjects and all coordinate planes, we find that the shape and orientation of perceptual isodiscrimination contours (red ellipses) are predicted by the distribution of image statistics extracted from natural scenes (blue ellipses). As in *Figure 3A*, the correspondence is very good, with mismatch that is comparable to the variability observed across image analyses and across subjects.

To quantify the correspondence between natural image and psychophysical analyses, we describe each ellipse by a single vector $\vec{\omega}$ that combines information about shape (eccentricity) and orientation (angular tilt), and we compute the scalar product between the image analysis vector $\vec{\omega}_{NI}$ and the subject-averaged psychophysical vector $\vec{\omega}_{PP}$. This value, averaged across coordinate planes, ranges from 0.953 to 0.977 across image analyses. We compared this correspondence to that obtained under the null hypotheses that (*i*) the apparent correspondence between image statistic covariances and isodiscrimination contours is chance, or (*ii*) the apparent covariances in image statistics are due to chance. The observed correspondence is much greater than the value measured under either null hypothesis ($p \leq .0003$ for each image analysis under both hypotheses; see 'Materials and methods', *Analysis of image statistics in pairwise coordinate planes*, and *Figure 3—figure supplement 2* for comparisons of eccentricity and tilt, and *Tables 1–3* and 'Materials and methods', *Permutation tests*, for statistical tests).

These findings confirm that the shape and orientation of human isodiscrimination contours, measured across all pairwise combinations of coordinates, can be quantitatively predicted from the covariation of image statistics extracted from natural scenes. The observed correspondence is maintained within the full 9-dimensional coordinate space (see 'Materials and methods', *Analysis of the full 9-dimensional distribution of image statistics*, and *Figure 3—figure supplement 3* for principal component analyses, and *Tables 1–3* and 'Materials and methods', *Permutation tests*, for statistical tests), confirming that our hypothesis describes human sensitivity in the full 9-dimensional space of local image statistics extracted from natural scenes.

## Discussion

How should neural mechanisms be distributed to represent a diverse set of informative sensory features? We argued that, when performance requires inferences limited by sampling of the statistics of input features, resources should be devoted in proportion to feature variability. A basic idea here is that features that take a wider range of possible values are less predictable, and will better distinguish between contexts in the face of input noise. We used this hypothesis to successfully predict human sensitivity to elements of visual form arising from spatial multi-point correlations in images. This result is notable for several reasons. First, we successfully predicted dozens of independent parameters that describe human perceptual sensitivity. The only free parameter was a scale that converted between perceptual sensitivities and natural image statistics. Moreover, predictions about the rank ordering of sensitivities (*Figure 3A*) and the shape and orientation of isodiscrimination contours (*Figure 3B*) do not even require a scale factor. Second, the theoretical predictions and their psychophysical test were derived from two very different sources. Psychophysical stimuli consisted of mathematically-defined synthetic binary textures with precisely-controlled correlational structure that is unlikely to occur outside of the laboratory. In contrast, the efficient coding predictions were derived from calibrated

photographs of natural scenes in which many types of correlations are simultaneously present. Third, predictions refer to multi-point (and not just pairwise) correlations, which are critical for defining local features such as lines and edges (*Oppenheim and Lim, 1981*; *Morrone and Burr, 1988*). In contrast, previous normative theories have have mainly focused on explaining the linear receptive fields of neurons in primary visual (*Olshausen and Field, 1996*; *Bell and Sejnowski, 1997*; *van Hateren and Ruderman, 1998*; *van Hateren and van der Schaaf, 1998*; *Hyvarinen and Hoyer, 2000*; *Vinje and Gallant, 2000*; *Karklin and Lewicki, 2009*) and auditory cortex (*Carlson and DeWeese, 2002*, *2012*), or on deriving symmetry- and coverage-based mesoscopic models of cortical map formation in V1 (*Wolf and Geisel, 1998*; *Swindale et al., 2000*; *Kaschube et al., 2011*). Finally, the efficient coding prediction of greater sensitivity to more variable multipoint correlations is closely tied to the statistical structure of natural visual images. Specifically, this regime applies to highly variable multipoint correlations that cannot be predicted from simpler ones. Some other multipoint correlations (defined on configurations other than a 2 × 2 glider) are also highly variable, but they are predictable from simpler correlations. For these multipoint correlations, visual sensitivity is very low (*Tkačik et al., 2010*), and efficient coding is not applicable in the form proposed here.

In sum, the surprising predictive power and the high statistical significance of our results provide strong support for the proposed application of the efficient coding hypothesis to cortical processing of complex sensory features.

## Perceptual salience of multi-point correlations likely arises in cortex

Although we did not record cortical responses directly, several lines of evidence indicate that that the perceptual thresholds we measured are determined by cortical processes. First, the stimuli had high contrast (100%) and consisted of pixels that were readily visible (14 arcmin), so retinal limitations of contrast sensitivity and resolution were eliminated. Second, the task requires pooling of information over wide areas (100–200 pixels, that is, a region whose diameter is 10–15 times the width of an image element; see Figure 7 in *Victor and Conte, 2005*). Retinal receptive fields are unlikely to do this, as the ratio of their spatial extent (surround size) to their resolution (center size) is typically no more than 4:1 (*Croner and Kaplan, 1995*; *Kremers et al., 1995*). Third, to account for the specificity of sensitivity to three- and four-point correlations, a cascade of two linear-nonlinear stages is required (*Victor and Conte, 1991*); retinal responses are quite well-captured by a single nonlinear stage (*Nirenberg and Pandarinath, 2012*), and cat retinal populations show no sensitivity to the four-point correlations studied used here (*Victor, 1986*) while simultaneous cortical field potentials do. Conversely, macaque visual cortical neurons (*Purpura et al., 1994*), especially those in V2, manifest responses to three- and four-point correlations (*Yu et al., 2013*).

## Cortex faces a different class of challenges than the sensory periphery

Successive stages of sensory processing share the same broad goals: invest resources in encoding stimulus features that are sufficiently informative, and suppress less-informative ones. In the periphery, this is exemplified by the well-known suppression of very low spatial frequencies; in cortex, this is exemplified by insensitivity to high-order correlations that are predictable from lower-order ones. Previous work has shown that such higher-order correlations can be separated into two groups—informative and uninformative—and only the informative ones are encoded (*Tkačik et al., 2010*). We used this finding to select an informative subspace for the present study, and we asked how resources should be efficiently allocated amongst features within this informative subspace.

A simple model of efficient coding by neural populations is shown in *Figure 4A* (details in 'Materials ans methods', *Two regimes of efficient coding*). Here, to enable analytical calculations, we used linear filters of variable gain and subject to Gaussian noise to model a population of neural channels encoding different features. The optimal allocation of resources to maximize information transmitted by the population depends on the amount of input noise, the amount of output noise, the input signal variability, and the total resources available to the system, here quantified as a constraint on the total output power (i.e., sum of response variances) in the neural population. The constrained output power and the output noise together determine the 'bandwidth' of the system—that is, the expressive capacity of its outputs. Consider a neural population with input noise, output noise, and a fixed amount of output power. We find that when input signal variability is sufficiently large compared to the input noise, the gain of neurons should *decrease* with the variance of the input (regions to the right of the peaks in the right-hand panel of *Figure 4A*). This is a regime where the output bandwidth is low

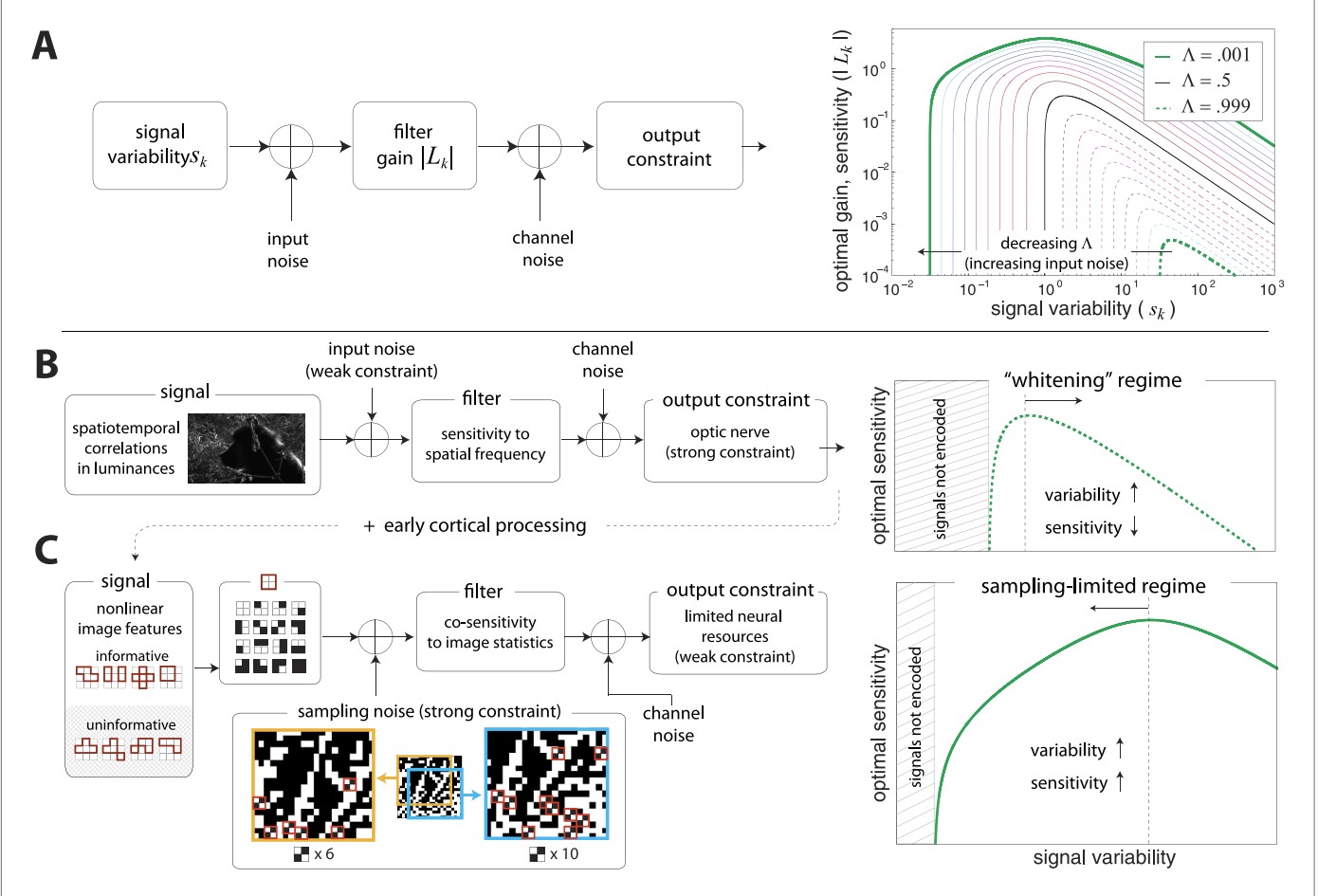

**Figure 4**. Regimes of efficient coding. (**A**) To analyze different regimes of efficient coding, we consider a set of channels, where the $k^{th}$ channel carries an input signal with variability $s_k$. Gaussian noise is added to the input. The result is passed through a linear filter with gain $|L_k|$, and then Gaussian noise is added to the filter output. We impose a constraint on the total power output of all channels, that is, a constraint on its total resources. With these assumptions, the set of gains that maximizes the transmitted information can be determined (see 'Materials and methods', *Two regimes of efficient coding*, and (*van Hateren, 1992a*; *Doi and Lewicki, 2011*; *Doi and Lewicki, 2014*)). This set of gains depends on the relative strengths of input and output noise and on the severity of the power constraint, quantified here by the dimensionless parameter $\Lambda$ (right-hand panel). As $\Lambda$ decreases from 1 to 0, the system moves from a regime in which output noise is limiting to one in which input noise is limiting. (**B**) The efficient coding model applied to the sensory periphery. Raw luminances from natural images are corrupted with noise (e.g. shot noise resulting from photon incidence) and passed through a linear filter. The resulting signal is carried by the optic nerve, which imposes a strong constraint on output capacity. In the bandwidth limited case where output noise dominates over input noise (e.g., under high light conditions when photon noise is not limiting), the optimal gain decreases as signal variability increases. Since channel input and channel gain vary reciprocally, channel outputs are approximately equalized, resulting in a 'whitening', or decorrelation. (**C**) The efficient coding model applied to cortical processing. Informative image features resulting from early cortical processing, caricatured by our preprocessing pipeline as applied to the retinal output, are sampled from a spatial region of the image. This sampling acts as a kind of input noise, because it only provides limited count-based estimates for the true statistical properties of the image source. When this input noise is limiting, the optimal gain *increases* as signal variability increases. Rather than whiten, the output signals preserve the correlational structure of the input. Note that in both regimes (**B**) and (**C**), there is a range of signals that are not encoded at all. These are the signals that are not sufficiently informative to warrant an allocation of resources.

The following figure supplements are available for figure 4:

**Figure supplement 1**. Schematic representation of channel optimization problem.

**Figure supplement 2**. Optimal coding regimes.

**Figure supplement 3**. Noise-dependent transition between efficient coding regimes.

**Figure supplement 4**. Optimal filter shape and orientation.

compared to the input range, and efficient coding predicts that signals should be 'whitened' by equalizing the variance in different channels. Conversely, consider input signals with a smaller range, which are thus more disrupted by input noise. In this case, the gain of neurons should *increase* with the variance of the input (regions to the left of the peaks in the right-hand panel of **Figure 4A**). This is a regime where the input noise dominates, and efficient coding predicts that the system should invest more resources in more variable, and hence more easily detectable, input signals. The relative sizes of input and output noise (controlled by $\Lambda$ in **Figure 4A**) determines the input ranges over which the two qualitatively different regimes of efficient coding apply.

To make these abstract considerations concrete, we first considered coding in the sensory periphery. A common strategy employed in the periphery is 'whitening', where relatively fewer resources are devoted (yielding lower gain) to features with more variation (**Olshausen and Field, 1996**). As an example, within the spatial frequency range that the retina captures well, sensitivity is greater for high spatial frequencies than for low ones, that is, sensitivity is inversely related to the degree of variation in natural scenes (the well-known $1/f^2$ power spectrum [**Olshausen and Field, 1996**]). **Figure 4B** illustrates how this strategy can emerge from the simple efficient coding scheme discussed above as applied to peripheral sensory processing. Spatiotemporal correlations of light undergo filtering before passing through the optic nerve bottleneck (a constraint on bandwidth). Such a constraint on bandwidth is equivalently understood as a regime where output noise is relatively large compared to input noise. In this limit, where output noise dominates over input noise, the optimal strategy is whitening (See **Srinivasan et al., 1982** and **Figure 4A**). Of course, real neural systems contend with both input and output noise; indeed recent work has shown that simply whitening to deal with output noise underestimates the optimal performance that the sensory periphery can achieve (**Doi and Lewicki, 2014**).

An alternative regime arises when input noise limits performance. In this regime, relatively *more* resources are devoted to features with more variation. This regime was discussed in early work of **van Hateren, (1992a)** and was also recognized in (**Doi and Lewicki, 2011**, **2014**), although it has received much less attention than the 'whitening' regime. Our results suggest that this is the regime is relevant to cortex, where it predicts the relative allocation of resources to higher-order image statistics. **Figure 4C** illustrates the simple efficient coding scheme in this context. We use our image preprocessing pipeline to mimic early visual processing, and we consider the downstream coding of higher-order image features. Because these features must be sampled from a finite patch of an image, they are subject to *input* noise arising from fluctuations in statistical estimation. When such input noise is limiting, the ability to detect a signal from noise increases with the variability of that signal. In this limit, efficient coding predicts that resources should be allocated in proportion to feature variability (**Figure 4C**). This captures the intuition that when signal reliability is in question, more reliable signals warrant more resources. Furthermore, if two or more channels have covarying signals, resources should be devoted in relation to the direction and degree of maximum covariance (see 'Materials and methods', *Two regimes of efficient coding*, **Figure 4—figure supplement 3**, and **Figure 4—figure supplement 4**).

The difference between these two efficient coding regimes is a consequence of the form of noise—output vs input noise—that is limiting. Our finding that cortex operates in a different regime than the well-known peripheral whitening reflects the fact that different stages and kinds of processing can face different constraints. While information transmission by the visual periphery is limited by a bottleneck in the optic nerve, cortex faces no such transmission constraint. Furthermore, while faithful encoding may be an immediate goal of early visual processing, cortical circuits have to interpret image features from a complex and crowded visual scene and perform statistical inference. For example, to discriminate between various textures, the cortex cannot perform pixel-by-pixel comparisons, but must rely on the estimation of local correlations (image statistics) instead. Because these correlations must be sampled from a finite patch of the visual scene, any estimate will be limited by sampling fluctuations.

## Sampling constraints vs resource constraints

Sampling fluctuations constitute a source of input noise, the magnitude of which depends on the size of the sampled region. For natural images, this gives rise to a tradeoff: small regions lead to large fluctuations in the estimated statistics, while large regions blur over local details. This blurring may obscure the boundaries between objects with different surface properties. While the brain must implement such sampling, the size, scale, and potentially dynamic nature of the sampling region is not known. Interestingly, our predictions of human sensitivities do not change substantially over a wide

range of spatial scales and image patch sizes, perhaps reflecting a scaling property of natural images (*Stephens et al., 2013*). An avenue for future research is to determine whether there is an optimal region size, and if so, whether it could be estimated from images themselves.

Sampling limitations alone do not suffice to account for the observed differential sensitivity of the brain to local image statistics. Were sampling limitations the only consideration, perceptual sensitivity would be the same along each coordinate axis, and perceptual isodiscrimination contours would be circular in each coordinate plane. This follows from an ideal observer calculation (See Appendix B of *Victor and Conte, 2012*). In contrast, we find that human observers have a severalfold variability in sensitivity along different coordinate axes (*Figure 3A*) and have isodiscrimination contours that are elongated in oblique directions (*Figure 3B*). The efficient coding principle can account for these findings by taking into consideration the fact that a real observer has finite processing resources. In this context (finite resources *and* substantial input noise), the efficient coding principle predicts that resources are invested in relation to the range of signal values that are typically present (*van Hateren, 1992a*), as we find. Interestingly, resource limitations seem to play an important role in the cortex despite the vast expansion in the number of neurons compared to the optic nerve. Presumably, this reflects the large number of complex features that could be computed and the corresponding need for a large overrepresentation of the stimulus space (*Olshausen and Field, 1997*).

## Clues to neural mechanisms

While we find a close match between the variation in natural image statistics and human psychophysical performance, some aspects of the distribution of natural image statistics do not match psychophysical data.

These differences are not readily apparent when we examine the variances and covariances (*Figure 3*) of the distribution of natural image statistics but emerge only when one considers its detailed shape (see 'Materials and methods', *Asymmetries in distributions of natural image statistics*). For example, the distribution of $\alpha$-coordinate values has a longer tail in the positive vs negative direction (see *Figure 3—figure supplement 9* and (*Tkačik et al., 2010*)). In contrast, human perceptual sensitivity is symmetric, or very nearly so (within ~ 20%), for positive vs negative values of $\alpha$ (*Victor et al., 2005*; *Victor and Conte, 2012*; *Victor et al., 2013*). This suggests that limitations imposed by 'neural hardware' force the system to use heuristics instead of matching the natural image distribution exactly. For example, an opponent mechanism responsible for detecting variations along, example, the $\alpha$ coordinate, might be a useful and easy (although imperfect) way to process the asymmetric distribution of four-point correlations found in natural scenes. Such a mechanism could be matched to the variance of the natural image distribution along the $\alpha$ coordinate, but not to its skew or other odd moments. An opponent mechanism would necessarily give rise to equal sensitivities to positive vs negative values of $\alpha$, as observed in psychophysical results. Further study of deviations from a perfect match to the distribution of natural image statistics might provide additional insight into these or other possible neural mechanisms, and into the goals of the computations. Independently, our results also raise an interesting theoretical question about the optimal representation of non-gaussian, multi-dimensional signals under resource-limited conditions.

## Outlook

Looking forward, we hypothesize that the principle of efficient coding might apply to cortical processing at higher levels. For example, more complex image features, such as shapes, are represented as conjunctions of contour fragments (*Brincat and Connor, 2004*), where each contour fragment is a local image object defined by particular multi-point correlations. We might speculate that the joint statistics of contour fragments in natural scenes can predict, through appropriate formulation of the same efficient coding principle used here, the properties of neurons in area IT (*Hung et al., 2012*; *Yau et al., 2012*) or the associated perceptual sensitivities of human observers.

Finally, although we have focused on perception of image statistics, we do this with the premise that this process is in the service of inferring the materials and objects that created an image and ultimately, guiding action. Thus, it is notable that we found a tight correspondence between visual perception and natural scene statistics without considering a specific task or behavioral set. Indeed, the emergence of higher-order percepts without explicit task specification was the original hope of the efficient coding framework as first put forward by Barlow and Attneave (*Attneave, 1954*; *Barlow, 1959*, *1961*). Doubtless, these 'top-down' factors also influence the visual computations

that underlie perception, and the nature and site of this influence are an important focus of future research.

## Materials and methods

### Image preprocessing

#### UPenn Natural Image Database

A database of images was collected in the Okavango Delta, a savannah habitat of Botswana (*Tkačik et al., 2011*). Panoramic, eye-level shots were taken with a Nikon D70 camera during the dry season in midday illumination. Trichromatic images were then converted to equivalent log-luminance images. From this database, we selected a set of 924 images with minimal amounts of sky (see following paragraph).

#### Image selection

Natural images were taken from two different databases: the UPenn Natural Image Database (shown *Figures 1 and 3*) and the van Hateren Natural Image Dataset (shown in 'Materials and methods', *Comparison with van Hateren Database*). Images from the UPenn Natural Image Database were selected by hand to ensure that they contained no man-made objects. We required that images contained minimal (less that one-third of the total image area) amounts of sky, as the contribution of sky to the overall power spectrum of natural images is well-documented (*Torralba and Oliva, 2003*) and is not the focus of the present study. Images from the van Hateren Natural Image Dataset were chosen subject to the additional constraint that scenery which was clearly the result of human landscaping (e.g. trees all in a line) be excluded. The analyses presented here were performed using the logarithms of the pixel intensities, a standard procedure in the study of natural images (*Ruderman and Bialek, 1994*). However, the results were unchanged if absolute pixel intensities were used instead. For more details about the construction of the images from these sources, see (*Tkačik et al., 2011*) (UPenn dataset) and http://www.kyb.tuebingen.mpg.de/?id=227 (van Hateren dataset).

#### Block averaging

Images of size $L_1 \times L_2$ are block-averaged by a factor of $N$, which involves averaging the intensities of pixels arranged into contiguous $N \times N$ squares. The resulting image is of size $L_1/N \times L_2/N$. To the extent that natural images are scale invariant (a well-supported hypothesis (*Field, 1987*; *Ruderman, 1997*; *Stephens et al., 2013*)), this procedure leaves the underlying statistics invariant. In our analyses, we block average images by at least a factor of two (thereby eliminating the Nyquist frequencies) in order to avoid sampling artifacts imposed by the camera matrix during image acquisition. In *Figures 1 and 3*, we presented two values of $N$: $N = 2, 4$. In 'Materials and methods', *Analysis variants for Penn Natural Image Database*, we show that our results are consistent when $N$ is extended to include $N = 8, 12, 16, 20$.

#### Fourier whitening

We divide each block-averaged image into square $R \times R$ patches. In *Figures 1 and 3*, we presented results using three values of $R$: $R = 32, 48, 64$. In 'Materials and methods', *Analysis variants for Penn Natural Image Database*, we show that our results are consistent when $R$ is extended to include $R = 80, 128$.

To remove global correlations in natural images, we whiten the set of image patches by flattening the Fourier power spectrum of the image patch ensemble. This procedure removes expected ensemble-average (and thus predictable) pairwise correlations, but non-zero pairwise correlations may still exist within individual patches; such correlations are the subject of this study. To carry out this procedure, the whitening filter is the inverse square-root of the ensemble-averaged Fourier power spectrum. For the natural image analyses presented here, the filter has a center-surround structure similar to that observed in the retina.

Following the whitening procedure, we binarize each image patch about its median pixel intensity. This creates image patches with equal numbers of black and white pixels.

#### Removal of blurry images

In any image database, there will be blurring due to camera motion and focus artifacts. Because we are interested in the statistics computed from in-focus image patches, we use a mixture of components (MOC) method to separate blurred from in-focus image patches.

To perform this separation, we first examined the 9-dimensional distribution of natural image statistics (see *Figure 1—figure supplement 1A* for the projection of the distribution onto the (α, β_) plane). When projected onto various coordinate planes, the structure of the distribution suggested that the distribution could be well-described by a weighted sum of two components. We explored this two-component description by running a standard maximum likelihood MOC inference that described each component by a Gaussian distribution. This inference method returned the mean, covariance, and relative weighting of each putative Gaussian component. In this process, each image patch was assigned to one of the two components (*Figure 1—figure supplement 1B*; note that the two components are separated in the 9-dimensional space, although they appear overlapping in this particular projection). After inspecting the clustering of patches into each of the two components, we observed that one of the components contained image patches that are sharp (*Figure 1—figure supplement 1C*), while the other contained patches that are blurry (*Figure 1—figure supplement 1D–E*).

We performed several controls to show that this separation is precise and effective. We first confirmed, based on visual inspection of a large number of images, that this method reliably separates blurred from in-focus patches. For example, images that were uniformly composed of patches assigned to the 'blurry' component were fully blurred due, example, to camera motion (*Figure 1—figure supplement 1E*). Similarly, images in which a large percentage of patches were assigned to the 'blurry' component contained large regions that were blurred due to motion or camera focus artifacts (*Figure 1—figure supplement 1D*). Furthermore, the spatial boundary between blurred and in-focus regions in the original image matched the boundary between patches assigned to the 'blurry' vs 'in-focus' component.

We additionally tested this method by incrementally removing images that were significantly blurred and then re-running the MOC method. After the removal of each subsequent image, the MOC method returned a mixture of components that was incrementally more strongly weighted toward the 'in-focus' component.

Finally, we tested this method by applying motion and Gaussian blur filters to sharp images (*Figure 1—figure supplement 2B*). With a sufficiently strong blurring transformation, all of the patches within a sharp image changed assignment from the 'in-focus' to the 'blurry' component. Successive block averaging removes the effects of small blur, such that a larger blurring transformation is required to change the assignment of patches from the 'in-focus' to the 'blurred' component. Furthermore, the application of motion and blur filters altered the spatial distribution of natural image statistics in a manner consistent with the statistics observed in image patches assigned to the 'blurry' component via the MOC method (*Figure 1—figure supplement 2A*). Both types of blurring increased the values of second- and fourth-order statistics, but they did so in different manners. Camera motion strongly increased both the fourth-order statistic and the second-order statistic aligned parallel to the direction of motion. In comparison, camera focus artifacts (arising, e.g., from variations in field of depth) more uniformly increased all second- and fourth-order statistics.

## Scaling image analyses

To compare between natural image and psychophysical analyses, we scale the set of 9 standard deviations extracted from a given image analysis by a multiplicative factor that minimizes the squared error between the set of nine standard deviations and the set of nine psychophysical sensitivities. *Figure 3—figure supplement 1* shows the value of the scale factor for different choices of the block-average factor $N$ and patch size $R$. This scaling places the greatest weight on the match between statistics with high variation/sensitivity (i.e. $\beta_|$ and $\beta_-$). Note that a different choice of scaling factor can shift this weight to different statistics; for example, a scaling factor that minimizes the least squares error between inverse standard deviation and thresholds will place larger weight on the match between statistics with low variation/sensitivity (i.e. $\theta$ components).

## Psychophysical methods

We determined perceptual sensitivity to local image statistics via a texture segmentation paradigm adapted from (*Chubb et al., 2004*), and in standard use in our lab (*Victor et al., 2005*; *Victor and Conte, 2012*; *Victor et al., 2013*); we describe it briefly here. These measurements were carried out in parallel with the natural scene analysis described above. Some of the psychophysical results have been previously reported (*Victor and Conte, 2012*; *Victor et al., 2013*); see 'Subjects' below.

## Stimuli

The basic stimulus consisted of a 64 × 64 black-and-white array of square image elements ('checks'), in which a target 16 × 64 rectangle of checks was embedded, positioned eight checks from one of the four edges of the array. The target was distinguished from the rest of the array by its local statistical structure (see *Victor and Conte, 2012* for details on the synthesis of these images), which was varied as described below.

Individual experimental sessions consisted of threshold measurements for each of a pair of image statistic coordinates (i.e., two choices from $\{\beta_|, \beta_-, \beta_\backslash, \beta_/, \theta_\ulcorner, \theta_\urcorner, \theta_\lrcorner, \theta_\llcorner, \alpha\}$), and their pairwise interactions. For the trials used to determine the sensitivity along a coordinate axis, the coordinate was set to one of five equally spaced values; lower-order coordinates were set to 0, and higher-order coordinates were set to their maximum-entropy values (0 for all cases except the $(\beta, \alpha)$ pair; see (*Victor and Conte, 2012*) for further details on this point). The highest coordinate value tested was determined from pilot experiments, and was set at 0.45 for $\beta_|$ and $\beta_-$, 0.75 for $\beta_\backslash$ and $\beta_/$, 1.0 for the $\theta$'s, and 0.85 for $\alpha$. For the trials used to determine the sensitivity to pairwise combinations of coordinates, each coordinate was given a nonzero value; all sign combinations were used. The ratio of the coordinate magnitudes was fixed, and chosen in approximate proportion to the above maximum values. Two values for each sign combination were studied.

To ensure that the response was driven by figure/ground segmentation (rather than, say, a texture gradient), two kinds of trials were randomly intermixed: (1) trials in which the target contained the nonzero value(s) of the coordinates and the background was random (i.e., all coordinates set to 0), and (2) trials in which the background had the nonzero values, and the target was random. Targets were equally likely to appear in any of the four possible locations. All trials were intermixed. This amounted to a total of 288 trials per block along eight rays. We collected 15 such blocks per subject (4320 trials) for each coordinate pair, and tested 11 pairs, for a total of 47,520 trials per subject: $(\beta_-, \beta_|)$, $(\beta_-, \beta_\backslash)$, $(\beta_\backslash, \beta_/)$, $(\beta_-, \theta_\lrcorner)$, $(\beta_\backslash, \theta_\lrcorner)$, $(\beta_/, \theta_\lrcorner)$, $(\theta_\lrcorner, \theta_\llcorner)$, $(\theta_\lrcorner, \theta_\urcorner)$, $(\beta_-, \alpha)$, $(\beta_\backslash, \alpha)$, $(\theta_\lrcorner, \alpha)$. These pairs encompass all the distinct coordinate pairs, up to 4-fold rotational symmetry. Since there was no detectable dependence on the orientation of pairwise or third-order correlations related by rotational symmetry in pilot experiments, measurements along coordinate axes and coordinate planes related by rotation are pooled in *Figure 3* and in *Figure 3—figure supplements 5–8*.

Stimuli were presented on a mean-gray background, followed by a random mask. The display was an LCD monitor with control signals provided by a Cambridge Research ViSaGe system; mean luminance of 23 cd/m² and refresh rate was 100 Hz. The stimulus size was 15° × 15° (check size of 14 min), contrast was 1.0, and viewing distance was 1m. Presentation time was 120 ms.

## Subjects

Four normal subjects (2 male, 2 female), ages 23 to 54 participated. One subject (MC) was a very experienced observer (several thousand hours); the other three had modest viewing experience (10–100 hr) prior to the experiment. JD and DF were naive to the purposes of the experiment. All subjects had visual acuities (corrected if necessary) of 20/20 or better. For subjects MC and DT, data from all coordinate planes other than the $(\beta_\backslash, \alpha)$-plane were previously reported (*Victor and Conte, 2012*; *Victor et al., 2013*). For subjects JD and DF, data from the seven pairs of coordinates not containing $\alpha$ were previously reported (*Victor et al., 2013*).

## Procedure

Subjects were asked to indicate the position of the target (4-alternative forced choice), by pressing one of four buttons. They were informed that the target was equally likely to appear in any of four locations (top, right, bottom, left), and were shown examples of stimuli of both types (target structured/background random and target random/background structured) prior to the experiment. Subjects were instructed to fixate centrally and not scan the stimulus. During training but not data collection, auditory feedback for incorrect responses was given. After performance stabilized (approx. 3 hrs for a new subject), data collection began. Within blocks, trial order was random. Block order was counterbalanced across subjects.

## Determination of sensitivity

To summarize the psychophysical performance, we fit Weibull functions to the fraction correct (FC) for each subject and each kind of block (i.e., each pair of coordinates). In the first step of the analysis

of each dataset, maximum-likelihood fits were obtained separately for each of its eight rays $r$ (the rays consisted of the positive and negative values for the two coordinates, and the four diagonal directions):

$$FC(x) = \frac{1}{4} + \frac{3}{4}\left(1 - 2^{-(x/a_r)^{b_r}}\right),$$ (0.1)

where $x$ is the Euclidean distance from the coordinate vector to the origin, $a_r$ is the distance at which FC = 0.625 (halfway between chance and perfect), and $b_r$ is a shape parameter, controlling the slope of the psychophysical curve. Since the shape parameter $b_r$ was usually in the range 2.2–2.7 for each pairwise coordinate plane, we then fit the entire dataset within each plane by a set of Weibull functions constrained to share a common exponent $b$, but allowing the parameter $a_r$ to vary across rays. For each on-axis ray, we averaged the value of $1/a_r$ obtained from all planes that included the ray (these were mutually consistent (*Victor et al., 2013*)) to obtain a final value for the perceptual sensitivity.

## Determination of isodiscrimination ellipsoids

To determine the isodiscrimination ellipsoids, we first parameterized them by a quadratic $\sum_{i,j} Q_{ij} c_i c_j$, where $c_i$ and $c_j$ each represent one of the local coordinates $\{\beta_|, \beta_-, \beta_\backslash, \beta_/, \theta_\vdash, \theta_\dashv, \theta_⊥, \theta_⊤, \alpha\}$, and $Q_{ij}$ is the symmetric matrix for which criterion performance (FC = 0.625) is reached at $\sum_{i,j} Q_{ij} c_i c_j = 1$. The values of $Q_{ij}$ were obtained by minimizing:

$$F = \sum_r \left(\left(\sum_{i,j} Q_{ij} c_i(T_r) c_j(T_r)\right) - 1\right)^2,$$ (0.2)

where $T_r$ is the texture along the ray $r$ at which criterion performance is reached (i.e., the texture at a distance $a_r$ from the origin, where $a_r$ is the sensitivity along the ray $r$, as determined above), and $c_{ij}(T)$ is the value of the $i$th coordinate for the texture $T_r$. This minimization is a linear least-squares procedure in the $Q_{ij}$. Deviation of the fitted values of $\sum_{i,j} Q_{ij} c_i(T_r) c_j(T_r)$ from unity, which corresponds to deviation of the fitted ellipsoidal surface from the measured points of criterion performance, ranged from 7–10% (root-mean-squared) across subjects. The ellipses shown in *Figure 3B*, *Figure 3—figure supplement 6*, and *Figure 3—figure supplement 7* correspond to loci at which $\sum_{i,j} Q_{ij} c_i c_j$ is constant, and the eigenvectors described in *Figure 3—figure supplements 3* and *8* are the eigenvectors of $Q$.

## Analysis of image statistics in pairwise coordinate planes

In pairwise coordinate planes, our hypothesis predicts that the inverse covariance matrix, or precision matrix, matches human isodiscrimination contours. A precision matrix is represented by the contour lines of its inverse (the covariance matrix $M$); these are the points $(x, y)$ at which $M_{xx}x^2 + 2M_{xy}xy + M_{yy}y^2 =$ constant. A short distance of this contour from the origin thus indicates a large value of $M$ and a small value of the precision matrix. This in turn denotes a direction in which prior knowledge of the image statistic is imprecise.

*Figure 3B* shows a correspondence between contours of the precision matrix (extracted from natural images) and human isodiscrimination contours. This is shown again here in *Figure 3—figure supplement 2A* for subject-specific (lower half grid) and subject-averaged (upper half grid) isodiscrimination contours. This correspondence can be made quantitative by computing the angular tilt (*Figure 3—figure supplement 2B*) and eccentricity (*Figure 3—figure supplement 2C*) of each ellipse. Across all 36 pairwise coordinate planes, we find a detailed quantitative match between the shape and orientation of precision matrix contours and human isodiscrimination contours.

## Analysis of the full 9-dimensional distribution of image statistics

### Principal component decomposition

Here, we verify our hypothesis within the full 9-dimensional space of image statistics using an approach that does not single out coordinate axes, either individually or in pairs. Just as the projections of the natural image distribution can be fit by a bivariate Gaussian in each coordinate plane, the entire distribution can be fit by a multivariate Gaussian in the full 9-dimensional space. Similarly,

the full set of perceptual isodiscrimination contours can be fit by a single 9-dimensional ellipsoid. Our hypothesis predicts that these two 9-dimensional ellipsoids have the corresponding shape and orientation.

To test this, we compare the principal axes $\{\vec{\xi}_{NI}\}$ of variation in natural scenes with the principal axes $\{\vec{\xi}_{PP}\}$ of human sensitivity inferred from the ellipsoidal isodiscrimination surface (*Victor et al., 2013*). To aid in this comparison, we first align the two sets of principal axes based on eigenvalue rank and symmetry considerations (discussed below). We then compute the fractional contribution $f$ of sets of coordinates to each principal axis $\vec{\xi}^{(i)}$, therein grouping coordinates with similar ranges of variation. *Figures 3—figure supplement 3A–D* respectively show the fractional contributions $f_{\{\beta_|,\beta_-\}}$, $f_{\{\beta_\backslash,\beta_/\}}$, $f_{\{\theta_\llcorner,\theta_\lrcorner,\theta_\ulcorner,\theta_\urcorner\}}$, and $f_\alpha$ to $\{\vec{\xi}_{NI}\}$ (blue bars) vs $\{\vec{\xi}_{PP}\}$ (red bars).

We find that the principal axes of variation in natural scenes match the principal axes of human sensitivity. As observed in *Figure 3*, the correspondence is within the range of variability observed across image analyses and human subjects.

We quantify the overlap between each image analysis and the set of psychophysical analyses by computing the scalar product between each principal component vector $\vec{f}_{NI}^{(i)}$ extracted from natural images and the corresponding subject-averaged psychophysical vector $\vec{f}_{PP}^{(i)}$, where $\vec{f} = \{f_{\beta_|}, f_{\beta_\backslash}, f_\theta, f_\alpha\}$. This overlap, averaged across principal components, ranges from 0.991 to 0.996 across image analyses and is consistently larger than the overlap measured under null hypotheses in which patch labels and coordinate labels are independently shuffled ($p \leq .0004$ for each image analysis under both hypotheses; see Appendix 4 for details).

## Alignment of principal components

As described in the previous subsection, we use principal component analysis for the multivariate comparison of natural image statistics and perceptual sensitivities. In addition to the standard approach of ordering components by percentage of variance explained within each dataset, followed by comparing components of corresponding rank, we use an additional tool: the symmetries in the definitions of the image statistic coordinates. As detailed below, we use these symmetries to group principal components into symmetry classes, and we then rank-order the components within each class. By matching components based on both symmetry and rank order of explained variance, we avoid ambiguities that would otherwise occur if only explained variance was considered. The four symmetry classes are defined as follows:

1. 4-D subspace in which statistics are invariant under 90° rotations in the plane (here, designated 'SYM'). This is spanned by:

(i) $\beta_| = \beta_-$, all else 0 $\left(\left[\frac{1}{\sqrt{2}}, \frac{1}{\sqrt{2}}, 0, 0, 0, 0, 0, 0, 0\right]\right)$

(ii) $\beta_\backslash = \beta_/$, all else 0 $\left(\left[0, 0, \frac{1}{\sqrt{2}}, \frac{1}{\sqrt{2}}, 0, 0, 0, 0, 0\right]\right)$

(iii) $\theta_\ulcorner = \theta_\urcorner = \theta_\lrcorner = \theta_\llcorner$, all else 0 $\left(\left[0, 0, 0, 0, \frac{1}{2}, \frac{1}{2}, \frac{1}{2}, \frac{1}{2}, 0\right]\right)$

(iv) $\alpha \neq 0$, all else 0 ($[0,0,0,0,0,0,0,0,1]$)

2. 2-D subspace in which coordinate values are negated after a horizontal or vertical mirror (here, designated 'HVI'). This is spanned by:

(i) $\beta_\backslash = -\beta_/$, all else 0 $\left(\left[0, 0, \frac{1}{\sqrt{2}}, -\frac{1}{\sqrt{2}}, 0, 0, 0, 0, 0\right]\right)$

(ii) $\theta_\ulcorner = -\theta_\urcorner = \theta_\lrcorner = -\theta_\llcorner$, all else 0 $\left(\left[0, 0, 0, 0, \frac{1}{2}, -\frac{1}{2}, \frac{1}{2}, -\frac{1}{2}, 0\right]\right)$

3. 2-D subspace spanned by two vectors $v_1$ and $v_2$ for which a 90° rotation transforms $v_1$ to $v_2$ and $v_2$ to $-v_1$ (here, designated 'ROT'). This is spanned by:

(i) $\theta_\ulcorner = -\theta_\lrcorner$, all else 0 $\left(\left[0, 0, 0, 0, \frac{1}{\sqrt{2}}, 0, -\frac{1}{\sqrt{2}}, 0, 0\right]\right)$

(ii) $\theta_\urcorner = -\theta_\llcorner$, all else 0 $\left(\left[0, 0, 0, 0, 0, \frac{1}{\sqrt{2}}, 0, -\frac{1}{\sqrt{2}}, 0\right]\right)$

4. 1-D subspace in which a diagonal mirror negates coordinates (here, designated "DII"). This is spanned by:

(i) $\beta_| = -\beta_-$, all else 0 $\left(\left[\dfrac{1}{\sqrt{2}}, -\dfrac{1}{\sqrt{2}}, 0, 0, 0, 0, 0, 0, 0\right]\right)$

We compute the normalized principal axes $\{\bar{\xi}_{NI}\}$ of variability in natural image statistics and principal axes $\{\bar{\xi}_{PP}\}$ of human perceptual sensitivity. We then assign each set of components to the above symmetry classes by maximizing the total overlap between $\{\bar{\xi}\}$ and the above classes. This is accomplished by computing the size of the projection of each individual component $\bar{\xi}^{(i)}$ into each of the above subspaces, and then assigning the component into the subspace that contains the largest projection. In one case where two components with nearly degenerate eigenvalues could not clearly be assigned to symmetry classes (analysis N = 20, R = 32 in the PIDB, shown in *Figure 3—figure supplement 8A–D* below), we force symmetry by performing a 45° rotation in the plane spanned by the degenerate components.

Once all components have been assigned to symmetry classes, we rank-order components within each class. This resulted in unambiguous pairing between natural image dataset and psychophysics in all but one pair of components in three image analyses (out of a total of 9 components for each of 31 separate image analyses). In those analyses (image analyses N = 2, R = 48, 64, 128 in the van Hateren database), there were two nearly-degenerate SYM components in the image dataset; we paired these components with the psychophysics data by maximizing their overlap.

To compare between natural image and psychophysics analyses, we compute the fractional contribution $\vec{f}^{(i)} = [f_{\beta_{|-}}^{(i)}, f_{\beta_{\backslash /}}^{(i)}, f_{\theta}^{(i)}, f_{\alpha}^{(i)}]$ of sets of coordinates to each principal component, where the components of $\vec{f}^{(i)}$ are given by:

$$f_{\beta_{|-}}^{(i)} = \left(\xi_{\beta_|}^{(i)}\right)^2 + \left(\xi_{\beta_-}^{(i)}\right)^2 \tag{0.3}$$

$$f_{\beta_{\backslash /}}^{(i)} = \left(\xi_{\beta_\backslash}^{(i)}\right)^2 + \left(\xi_{\beta_/}^{(i)}\right)^2 \tag{0.4}$$

$$f_{\theta}^{(i)} = \left(\xi_{\theta_\ulcorner}^{(i)}\right)^2 + \left(\xi_{\theta_\urcorner}^{(i)}\right)^2 + \left(\xi_{\theta_\llcorner}^{(i)}\right)^2 + \left(\xi_{\theta_\lrcorner}^{(i)}\right)^2 \tag{0.5}$$

$$f_{\alpha}^{(i)} = \left(\xi_{\alpha}^{(i)}\right)^2 \tag{0.6}$$

and $f_{\beta_{|-}}^{(i)} + f_{\beta_{\backslash /}}^{(i)} + f_{\theta}^{(i)} + f_{\alpha}^{(i)} = 1$ for each normalized component $\bar{\xi}^{(i)}$.

The principal components shown in *Figure 3—figure supplement 3* are rank-ordered within each symmetry class, where the four classes were ordered as follows: SYM ($\bar{\xi}^{(1)} - \bar{\xi}^{(4)}$), HVI ($\bar{\xi}^{(5)}$, $\bar{\xi}^{(6)}$), ROT ($\bar{\xi}^{(7)}$, $\bar{\xi}^{(8)}$), DII ($\bar{\xi}^{(9)}$). Note that while the comparisons between psychophysics and natural images are based on the squares of the principal components coordinates (*equations 0.3–0.6*) and is insensitive to their signs, the classification of principal components by symmetry classes guarantees that we are only comparing psychophysical and natural-image components for which the signs within each coordinate set ($\{\beta_|, \beta_-\}$, $\{\beta_\backslash, \beta_/\}$, and $\{\theta_\ulcorner, \theta_\urcorner, \theta_\llcorner, \theta_\lrcorner\}$) covary in the same fashion.

## Permutation tests

Our results, shown in *Figure 3* for single coordinates and pairwise coordinate planes, and extended to the full 9-dimensional distribution in *Figure 3—figure supplement 3*, show a consistent match between the variation in natural image statistics and psychophysical sensitivities. We quantify this match by first assigning vectors to the quantities shown in *Figure 3* and *Figure 3—figure supplement 3*, and then computing the overlap between natural image vectors and the corresponding psychophysical vectors. We consider the following vector quantities:

1. Single coordinates: We describe the range of variation in natural image statistics by the normalized 9-component vector of standard deviations $\bar{\sigma}_{NI} / \|\bar{\sigma}_{NI}\|$, where $\|\bar{v}\|$ denotes the L2 norm $\dfrac{1}{N}\sum_{i=1}^{N} v_i^2$ of a vector $\bar{v}$. Similarly, we describe the set of perceptual sensitivities by the normalized vector

$\vec{s}_{\mathrm{PP}} / \left\| \vec{s}_{\mathrm{PP}} \right\|$. In both cases, the vector components are measured with respect to the coordinates $\left\{ \beta_{|}, \beta_{-}, \beta_{\backslash}, \beta_{/}, \theta_{\ulcorner}, \theta_{\urcorner}, \theta_{\lrcorner}, \theta_{\llcorner}, \alpha \right\}$.

2. Pairwise coordinate planes: We describe each ellipse by the unit vector $\vec{\omega}$ that is a combined measure of eccentricity ($\epsilon$) and tilt ($\delta$). We define $\vec{\omega}$ on one quarter of the unit sphere: $\vec{\omega} = \sin \alpha \cos \delta \, \hat{\mathbf{x}} + \sin \alpha \sin \delta \, \hat{\mathbf{y}} + \cos \alpha \, \hat{\mathbf{z}}$, where $\epsilon = \sin \alpha$ and $\cos \delta$ are defined on the interval [0,1] (the second follows from the 180° rotational symmetry of ellipses). Note that this definition of $\vec{\omega}$ captures the ellipse property that when $\epsilon = \sin \alpha = 0$ (circular ellipses), $\delta$ is not defined. See ***Figure 3—figure supplement 4*** for a schematic of this representation.

3. Principal components: We consider two related measures for describing principal components. As shown in ***Figure 3—figure supplement 3***, we describe each principal component $\left\{ \bar{\xi}^{(i)} \right\}$ by the normalized vector $\vec{f}^{(i)} / \left\| \vec{f}^{(i)} \right\|$, which measures the fractional contribution of sets of statistics to the principal components $\bar{\xi}^{(i)}$. For a more detailed comparison, we can similarly describe each principal component by the normalized vector $\vec{F}^{(i)} / \left\| \vec{F}^{(i)} \right\|$, where $\vec{F}^{(i)} = [f_{\beta_{|}}^{(i)}, f_{\beta_{-}}^{(i)}, f_{\beta_{\backslash}}^{(i)}, f_{\beta_{/}}^{(i)}, f_{\theta_{\ulcorner}}^{(i)}, f_{\theta_{\urcorner}}^{(i)}, f_{\theta_{\lrcorner}}^{(i)}, f_{\theta_{\llcorner}}^{(i)}, f_{\alpha}^{(i)}]$. This measures the fractional contribution of individual statistics (rather than sets of statistics) to the principal components $\bar{\xi}^{(i)}$.

For each vector quantity ($\vec{\sigma}$, $\vec{\omega}$, $\vec{f}$, and $\vec{F}$), we compute the scalar product between a given image analysis vector and the subject-averaged psychophysical vector. We then report the overlap values (scalar products) measured for the six image analyses considered ***Figures 1 and 3*** ($N$ = 2, 4 and $R$ = 32, 48, 64). In computing the scalar product between $\vec{\omega}_{\mathrm{NI}}$ and $\vec{\omega}_{\mathrm{PP}}$, we report the overlap averaged over all 36 pairwise coordinate planes. Similarly, in computing the overlap between $\vec{f}_{\mathrm{NI}}$ and $\vec{f}_{\mathrm{PP}}$ and between $\vec{F}_{\mathrm{NI}}$ and $\vec{F}_{\mathrm{PP}}$, we report the overlap averaged over all 9 principal components. Note that, for each vector $\vec{\sigma}$, $\vec{\omega}$, $\vec{f}$, and $\vec{F}$, the maximum overlap is 1.

We find that natural image analyses show consistently high overlap with the set of psychophysical results (see ***Tables 1–3***). The overlap, as measured across image analyses, ranges from 0.988 to 0.999 for single coordinates ($\vec{\sigma}$), from 0.953 to 0.977 for pairwise coordinate planes ($\vec{\omega}$), from 0.987 to 0.993 for fractional principal axes ($\vec{f}$), and from 0.829 to 0.917 for the full principal axes ($\vec{F}$). We test the significance of this overlap by comparing our results to the following two null models:

1A. Shuffled coordinate labels: sets of coordinates. This model (and model 1b) tests the null hypothesis that the apparent correspondence between image statistic covariances and isodiscrimination contours is chance. We examine the 23 permutations of the sets of coordinates $\left\{ \beta_{|-}, \beta_{\backslash /}, \theta, \alpha \right\}$. We apply these permutations to the psychophysical data, as human subjects are equally sensitive to coordinates within each set ($\left\{ \beta_{|}, \beta_{-} \right\}$, $\left\{ \beta_{\backslash}, \beta_{/} \right\}$ and all $\theta$'s). This shuffling creates a new set of subjects whose second-order cardinal, second-order oblique, third-order, and fourth-order coordinate values are randomly permuted (transforming the original vector $[\beta_{|-}, \beta_{\backslash /}, \theta, \alpha]$ into, example, the shuffled vector $[\beta_{\backslash /}, \theta, \beta_{|-}, \alpha]$). If the correspondence between quantities derived from image analysis and psychophysics is statistically significant, we expect that the shuffled vectors $\vec{\sigma}$, $\vec{\omega}$, $\vec{f}$, and $\vec{F}$ will show less overlap with the image analysis vectors than do the original psychophysical vectors (note that the limited number of permutations restricts the minimum $p$-value to be 0.04).

1B. Shuffled coordinate labels: individual coordinates. Here, we expand the test described in 1a to randomly shuffle the full set of coordinate labels $\{ \beta_{|}, \beta_{-}, \beta_{\backslash}, \beta_{/}, \theta_{\ulcorner}, \theta_{\urcorner}, \theta_{\lrcorner}, \theta_{\llcorner}, \alpha \}$. In an analogous manner to that described in 2A, we expect that the shuffled vectors $\vec{\sigma}$, $\vec{\omega}$, $\vec{f}$, and $\vec{F}$ will show less overlap with the image analysis vectors than do the original psychophysical vectors if the correspondence between quantities derived from image analysis and psychophysics is statistically significant.

2. Shuffled patch labels. This model tests the null hypothesis that the apparent covariances in image statistics are due to chance. For each coordinate, we randomly shuffle image patch labels. This shuffling creates a new set of null patches whose second-, third-, and fourth-order coordinate values are randomly drawn from a subset of the original image patches (e.g. a given null patch can be described by a $\beta_{/}$-value measured from patch $m$ but an $\alpha$ value measured from patch $n$). This shuffling destroys correlations between coordinate values measured within individual patches.

Note that this shuffling does not alter the range of variation measured along single coordinate axes and will therefore not alter the values of precision matrix ellipses measured along coordinate axes. As a result, this test is not applicable to $\vec{\sigma}$, which measures natural image variation and human sensitivities along individual coordinate axes. However, shuffling will destroy correlations along oblique directions in coordinate planes, thereby aligning each ellipse along a single coordinate axis. Note that the eccentricity of each ellipse (in, e.g., the A-B plane) is then trivially related to the ratio of variances $\sigma^2$ measured along the corresponding coordinate axes: $\epsilon = \sqrt{1 - \sigma_A^2 / \sigma_B^2}$. We therefore expect that this shuffling will most strongly affect the tilt and eccentricity in pairwise planes in which ellipses are oriented along oblique directions ($\beta_|\beta_-$, $\beta_\backslash\beta_/$, and $\theta\theta$ planes). Finally, in destroying correlations between pairs of coordinates, this shuffling creates a diagonal covariance matrix, such that principal components are aligned with single coordinate axes. If the correspondence between quantities derived from image analysis and psychophysics is statistically significant, we expect that the shuffled vectors $\vec{\omega}$, $\vec{f}$, and $\vec{F}$ will show less overlap with the psychophysical vectors than do the original image analysis vectors.

Each null model is constructed by randomly selecting permuted indices that independently shuffle coordinate labels for subject-averaged psychophysical data (Null Model 1) and independently shuffle image patch labels for a given statistic (Null Model 2). For null model 1a, we perform the full set of 23 non-identity permutations. For models 1B and 2, we perform 10,000 permutations.

For each permutation, we compute a set of shuffled vectors $\left\{\vec{\sigma}, \vec{\omega}, \vec{f}, \vec{F}\right\}$, and we measure the overlap (defined as the scalar product $\overline{(*)}_{NI} \cdot \overline{(*)}_{PP}$) between each shuffled vector and the corresponding subject-averaged psychophysical vector. Note that, when assigning shuffled principal components to symmetry classes, no hand-tuning was performed. However, as described previously, such hand-tuning was only applied to a very small fraction of components for select image analyses.

When repeated for many permutations, this procedure yields a distribution of shuffled overlap values against which we measure the significance of the true (observed) overlap. Significance values (p-values) are estimated by computing the fraction of permutations for which the shuffled overlap exceeds the true overlap.

We find that the original image analyses show significantly higher overlap with psychophysical data than do the analyses produced by either of the null models. Results are significant for each measure of overlap and for each of the six analyses presented in *Figures 1 and 3* ($p < 0.0005$, or as small as possible given the number of possible permutations, in all cases); see *Tables 1–3* for full results.

## Analysis variants for Penn Natural Image Database

In *Figures 1 and 3*, we reported results using image analyses with varying values of the block-average factor N ($N = 2, 4$) and patch size R ($R = 32, 48, 64$). In *Figure 1—figure supplement 3*, we show that the relative variation in different image statistics (first shown in *Figure 1E*) is not an artifact of our image analysis pipeline, as the pattern of variation is destroyed if white-noise image patches are instead used. In Figures 3–figure supplement 5-3–figure supplement 8, we show that the comparison between natural image and psychophysical analyses is consistent across a wider range of image preprocessing parameters: $N = 2, 4, 8, 12, 16, 20$ and $R = 32, 48, 64, 80, 128$. Note that sampling limitations restrict some combinations of N and R (e.g. for sufficiently large N, we must choose sufficiently small R to have a statistically significant number of image patches).

## Comparison with van Hateren Database

All analyses reported in Results and shown in *Figures 1 and 3* were performed on a set of images from the UPenn Natural Image Database (*Tkačik et al., 2011*). Here, we extend our analyses to a set of 2300 images from the van Hateren image database (*van Hateren and van der Schaaf, 1998*), using the same set of parameters used to analyze images from the UPenn database, with block-average factors $N = 2, 4, 8, 12, 16, 20$ and patch sizes $R = 32, 48, 64, 80, 128$. Note that we are able to perform a larger number of analyses (specific combinations of N and R) than was performed using the Penn database, as we have a larger selection of images and therefore do not face the same sampling limitations. Figures 3—figure supplement 5-3–figure supplement 8 confirm that our results are consistent across image databases.

**Table 1.** Permutation tests for null model 1a: shuffled coordinate labels

| Measures of overlap | Image analysis | | Observed overlap | Shuffled overlap Values | | | | Significance |
|---|---|---|---|---|---|---|---|---|
| | | | | Mean | std | min | max | |
| Range/Sensitivity $\vec{\sigma}_{NI} \cdot \vec{s}_{PP}$ | N = 2 | R = 32 | 0.999 | 0.859 | $0.9 \times 10^{-1}$ | 0.704 | 0.983 | <0.04 |
| | | R = 48 | 0.993 | 0.832 | $1.1 \times 10^{-1}$ | 0.651 | 0.978 | <0.04 |
| | | R = 64 | 0.987 | 0.809 | $1.1 \times 10^{-1}$ | 0.614 | 0.974 | <0.04 |
| | N = 4 | R = 32 | 0.998 | 0.825 | $1.1 \times 10^{-1}$ | 0.638 | 0.969 | <0.04 |
| | | R = 48 | 0.994 | 0.812 | $1.1 \times 10^{-1}$ | 0.646 | 0.990 | <0.04 |
| | | R = 64 | 0.991 | 0.794 | $1.1 \times 10^{-1}$ | 0.617 | 0.985 | <0.04 |
| Inverse Range/Threshold $\langle \vec{\omega}_{NI} \cdot \vec{\omega}_{PP} \rangle$ | N = 2 | R = 32 | 0.971 | 0.709 | $1.5 \times 10^{-1}$ | 0.508 | 0.924 | <0.04 |
| | | R = 48 | 0.969 | 0.692 | $1.6 \times 10^{-1}$ | 0.469 | 0.924 | <0.04 |
| | | R = 64 | 0.953 | 0.685 | $1.7 \times 10^{-1}$ | 0.450 | 0.913 | <0.04 |
| | N = 4 | R = 32 | 0.967 | 0.679 | $1.7 \times 10^{-1}$ | 0.447 | 0.908 | <0.04 |
| | | R = 48 | 0.975 | 0.632 | $1.5 \times 10^{-1}$ | 0.400 | 0.880 | <0.04 |
| | | R = 64 | 0.977 | 0.648 | $1.6 \times 10^{-1}$ | 0.411 | 0.894 | <0.04 |
| Fractional Principal Components $\vec{f}_{NI} \cdot \vec{f}_{PP}$ | N = 2 | R = 32 | 0.994 | 0.382 | $1.5 \times 10^{-1}$ | 0.160 | 0.657 | <0.04 |
| | | R = 48 | 0.995 | 0.485 | $1.2 \times 10^{-1}$ | 0.287 | 0.727 | <0.04 |
| | | R = 64 | 0.991 | 0.487 | $0.7 \times 10^{-1}$ | 0.372 | 0.632 | <0.04 |
| | N = 4 | R = 32 | 0.995 | 0.459 | $1.4 \times 10^{-1}$ | 0.238 | 0.732 | <0.04 |
| | | R = 48 | 0.996 | 0.444 | $1.0 \times 10^{-1}$ | 0.277 | 0.601 | <0.04 |
| | | R = 64 | 0.996 | 0.450 | $1.1 \times 10^{-1}$ | 0.279 | 0.614 | <0.04 |
| Full Principal Components $\langle \vec{F}_{NI} \cdot \vec{F}_{PP} \rangle$ | N = 2 | R = 32 | 0.917 | 0.316 | $1.3 \times 10^{-1}$ | 0.123 | 0.578 | <0.04 |
| | | R = 48 | 0.828 | 0.401 | $1.0 \times 10^{-1}$ | 0.228 | 0.611 | <0.04 |
| | | R = 64 | 0.911 | 0.363 | $0.7 \times 10^{-1}$ | 0.282 | 0.532 | <0.04 |
| | N = 4 | R = 32 | 0.882 | 0.376 | $1.2 \times 10^{-1}$ | 0.180 | 0.618 | <0.04 |
| | | R = 48 | 0.917 | 0.362 | $1.0 \times 10^{-1}$ | 0.201 | 0.520 | <0.04 |
| | | R = 64 | 0.919 | 0.357 | $1.0 \times 10^{-1}$ | 0.196 | 0.522 | <0.04 |

We separately permute the sets of coordinate labels $\{\beta_{\perp}, \beta_{\backslash}, \theta, \alpha\}$. We apply these permutations to the psychophysical data, therein examining all 23 non-identity permutations of the four labels. This shuffling significantly decreases the overlap between image analyses and psychophysical data. Results are significant across all six analyses considered in **Figures 1 and 3** (N = 2, 4 and R = 32, 48, 64). p-values, estimated as the fraction of permutations for which the shuffled overlap exceeds the true overlap, are less than 0.04 (the minimum value given 23 permutations) for each image analysis.

## Asymmetries in distributions of natural image statistics

We find systematic asymmetries in the distributions of natural image statistics when examined beyond their second moments. **Figure 3—figure supplement 9** shows the distributions of single coordinates for the image analysis N = 2, R = 32. All distributions are shifted toward positive coordinate values, and there is larger variation in positive vs negative coordinate values. We assess this asymmetry in natural image analyses by computing the ratio of the standard deviations measured along positive vs negative coordinate axes. We similarly assess asymmetry in psychophysical analyses by computing the ratio of human sensitivities to positive vs negative deviations of coordinate values. This comparison is shown in **Figure 3—figure supplement 9**. The mismatch provides potential clues for the neural mechanisms responsible for processing local image statistics (See Discussion).

## Two regimes of efficient coding

In this section, we illustrate how two contrasting regimes emerge from the efficient coding principle: (*i*) the well-known transmission-limited regime, in which 'whitening' is optimal, and (*ii*) the sampling-limited regime, which is the focus of this paper. To enable exact calculations of optimal behavior, we consider a simplified scenario, in which all signals and noises are Gaussian, and all filters are linear.

**Table 2.** Permutation Tests for null model 1b: shuffled coordinate labels

| Measures of overlap | Image analysis | | Observed overlap | Shuffled overlap Values | | | | Significance |
|---|---|---|---|---|---|---|---|---|
| | | | | Mean | std | min | max | |
| Range/Sensitivity $\vec{\sigma}_{NI} \cdot \vec{s}_{PP}$ | N = 2 | R = 32 | 0.999 | 0.806 | $6.8 \times 10^{-2}$ | 0.659 | 0.999 | 0.0003 |
| | | R = 48 | 0.993 | 0.775 | $7.7 \times 10^{-2}$ | 0.610 | 0.993 | <0.0001 |
| | | R = 64 | 0.987 | 0.762 | $8.0 \times 10^{-2}$ | 0.579 | 0.987 | <0.0001 |
| | N = 4 | R = 32 | 0.998 | 0.828 | $6.0 \times 10^{-2}$ | 0.707 | 0.998 | <0.0001 |
| | | R = 48 | 0.994 | 0.798 | $7.1 \times 10^{-2}$ | 0.660 | 0.994 | 0.0002 |
| | | R = 64 | 0.991 | 0.780 | $7.6 \times 10^{-2}$ | 0.630 | 0.991 | <0.0001 |
| Inverse Range/Threshold $\langle \vec{\omega}_{NI} \cdot \vec{\omega}_{PP} \rangle$ | N = 2 | R = 32 | 0.971 | 0.693 | $8.1 \times 10^{-2}$ | 0.499 | 0.972 | 0.0002 |
| | | R = 48 | 0.969 | 0.682 | $8.4 \times 10^{-2}$ | 0.476 | 0.969 | 0.0003 |
| | | R = 64 | 0.953 | 0.671 | $8.5 \times 10^{-2}$ | 0.446 | 0.954 | 0.0002 |
| | N = 4 | R = 32 | 0.967 | 0.696 | $7.6 \times 10^{-2}$ | 0.521 | 0.964 | <0.0001 |
| | | R = 48 | 0.975 | 0.692 | $8.0 \times 10^{-2}$ | 0.509 | 0.976 | 0.0002 |
| | | R = 64 | 0.977 | 0.689 | $8.2 \times 10^{-2}$ | 0.493 | 0.978 | 0.0003 |
| Fractional Principal Components $\langle \vec{f}_{NI} \cdot \vec{f}_{PP} \rangle$ | N = 2 | R = 32 | 0.994 | 0.592 | $1.2 \times 10^{-1}$ | 0.271 | 0.995 | 0.0003 |
| | | R = 48 | 0.995 | 0.604 | $1.3 \times 10^{-1}$ | 0.281 | 0.995 | 0.0004 |
| | | R = 64 | 0.991 | 0.591 | $1.2 \times 10^{-1}$ | 0.278 | 0.991 | 0.0003 |
| | N = 4 | R = 32 | 0.995 | 0.590 | $1.2 \times 10^{-1}$ | 0.218 | 0.995 | 0.0001 |
| | | R = 48 | 0.996 | 0.577 | $1.2 \times 10^{-1}$ | 0.251 | 0.996 | 0.0002 |
| | | R = 64 | 0.996 | 0.581 | $1.2 \times 10^{-1}$ | 0.266 | 0.996 | 0.0004 |
| Full Principal Components $\langle \vec{F}_{NI} \cdot \vec{F}_{PP} \rangle$ | N = 2 | R = 32 | 0.917 | 0.391 | $1.2 \times 10^{-1}$ | 0.100 | 0.927 | 0.0002 |
| | | R = 48 | 0.828 | 0.391 | $1.2 \times 10^{-1}$ | 0.086 | 0.856 | 0.0008 |
| | | R = 64 | 0.911 | 0.396 | $1.2 \times 10^{-1}$ | 0.120 | 0.953 | 0.0003 |
| | N = 4 | R = 32 | 0.882 | 0.381 | $1.2 \times 10^{-1}$ | 0.066 | 0.989 | 0.0003 |
| | | R = 48 | 0.917 | 0.380 | $1.2 \times 10^{-1}$ | 0.090 | 0.902 | <0.0001 |
| | | R = 64 | 0.919 | 0.387 | $1.2 \times 10^{-1}$ | 0.095 | 0.937 | 0.0004 |

We separately permute all nine coordinate labels $\{\beta_{|}, \beta_{-}, \beta_{\backslash}, \beta_{/}, \theta_{\sqcap}, \theta_{\sqcup}, \theta_{\llcorner}, \theta_{\lrcorner}, \alpha\}$. This shuffling, applied to the psychophysical data, significantly decreases the overlap between image analyses and psychophysical data. Results are significant across all six analyses considered in **Figures 1 and 3** (N = 2, 4 and R = 32, 48, 64). p-values, estimated as the fraction of permutations for which the shuffled overlap exceeds the true overlap, are less than 0.0005 for all image analyses.

We consider a set of channels dedicated to processing independent signals of varying sizes. The channels, which are indexed by k, are abstract and general. For example, each k can represent a different spatial or temporal frequency in the input, as in the traditional analysis of visual coding in the periphery. Here, we take the signal on each channel k to represent a complex image feature, that is the result of a specific local nonlinear transformation applied to the input image.

*Figure 4—figure supplement 1* shows the setup of a single channel dedicated to processing the signal $s_k$. Sampling noise, which is assumed to be identical for each channel, is added to this signal; without loss of generality, we can take its value to be unity. Note that for the parametrization of local image statistics used here, sampling noise is in fact identical for each parameter at the origin of the parameter space (see Equations B19-B20 in *Victor and Conte, 2012*).

The result is passed through a linear filter $L_k$, characterized by a gain $|L_k|$. The output of $L_k$ then has intrinsic channel noise added, and the total dynamic range of all channels is constrained. All channels are assumed to have the same intrinsic noise. Again, without loss of generality, we take this value to be unity (as any scale associated with this noise can be absorbed into an overall multiplier for the filters $L_k$ and the constraint on total dynamic range of the channels).

We seek to find the optimal set of gains $\{|L_k|\}$ that maximize the mutual information $\sum_k H_k$ between the signals $\{s_k\}$ and the channel input, subject to a constraint Q on total output power. Using a

**Table 3.** Permutation tests for null model 2: shuffled patch labels

| Comparisons | Image analysis | | Observed overlap | Shuffled overlap Values | | | | Significance |
|---|---|---|---|---|---|---|---|---|
| | | | | Mean | std | min | max | |
| Inverse Range/Threshold $\langle \vec{\omega}_{NI} \cdot \vec{\omega}_{PP} \rangle$ | N = 2 | R = 32 | 0.971 | 0.924 | $0.70 \times 10^{-3}$ | 0.921 | 0.926 | <0.0001 |
| | | R = 48 | 0.969 | 0.921 | $1.1 \times 10^{-3}$ | 0.917 | 0.925 | <0.0001 |
| | | R = 64 | 0.953 | 0.912 | $1.3 \times 10^{-3}$ | 0.908 | 0.917 | <0.0001 |
| | N = 4 | R = 32 | 0.967 | 0.919 | $1.7 \times 10^{-3}$ | 0.914 | 0.926 | <0.0001 |
| | | R = 48 | 0.975 | 0.922 | $1.9 \times 10^{-3}$ | 0.916 | 0.930 | <0.0001 |
| | | R = 64 | 0.977 | 0.924 | $2.8 \times 10^{-3}$ | 0.916 | 0.935 | <0.0001 |
| Fractional Principal Components $\langle \vec{f}_{NI} \cdot \vec{f}_{PP} \rangle$ | N = 2 | R = 32 | 0.994 | 0.806 | $9.1 \times 10^{-6}$ | 0.806 | 0.806 | <0.0001 |
| | | R = 48 | 0.995 | 0.806 | $8.3 \times 10^{-6}$ | 0.806 | 0.806 | <0.0001 |
| | | R = 64 | 0.991 | 0.806 | $3.7 \times 10^{-6}$ | 0.806 | 0.806 | <0.0001 |
| | N = 4 | R = 32 | 0.995 | 0.807 | $2.5 \times 10^{-4}$ | 0.806 | 0.809 | <0.0001 |
| | | R = 48 | 0.996 | 0.807 | $4.1 \times 10^{-4}$ | 0.806 | 0.810 | <0.0001 |
| | | R = 64 | 0.996 | 0.807 | $3.5 \times 10^{-4}$ | 0.806 | 0.810 | <0.0001 |
| Full Principal Components $\langle \vec{F}_{NI} \cdot \vec{F}_{PP} \rangle$ | N = 2 | R = 32 | 0.917 | 0.448 | $5.8 \times 10^{-2}$ | 0.406 | 0.596 | <0.0001 |
| | | R = 48 | 0.828 | 0.502 | $5.9 \times 10^{-2}$ | 0.408 | 0.675 | <0.0001 |
| | | R = 64 | 0.911 | 0.458 | $4.8 \times 10^{-2}$ | 0.407 | 0.591 | <0.0001 |
| | N = 4 | R = 32 | 0.881 | 0.489 | $4.9 \times 10^{-2}$ | 0.409 | 0.638 | <0.0001 |
| | | R = 48 | 0.917 | 0.454 | $3.0 \times 10^{-2}$ | 0.408 | 0.637 | <0.0001 |
| | | R = 64 | 0.919 | 0.492 | $4.2 \times 10^{-2}$ | 0.411 | 0.648 | <0.0001 |

Within each image analyses, we separately permute image patch labels along individual coordinate axes. This shuffling does not alter the range of variation observed along individual coordinates; as a result, this test only applies to $\vec{\omega}$, $\vec{f}$ and $\vec{F}$. We find that this shuffling significantly decreases the overlap between image analyses and psychophysical data. Results are significant across all six analyses considered in **Figures 1 and 3** (N = 2, 4 and R = 32, 48, 64). p-values, estimated as the fraction of permutations for which the shuffled overlap exceeds the true overlap, are less than 0.0001 for each image analysis.

Lagrange multiplier $\Lambda$ for the constraint, the problems translates into extremizing $P = \sum_k H_k + \Lambda Q$ by setting $\partial P / \partial L_k = 0$.

The solution can be found in Equation 8 of **van Hateren, (1992a)**, noting the following correspondences between the setup of **Figure 4—figure supplement 1** and the scenario considered in that paper. Referring to the notation in (**van Hateren, 1992a**), the input and channel noises, $N_p$ and $N_c$, respectively correspond here to the sampling and channel noises (both taken to be unity). The prefiltered stimulus power $S_p$ corresponds here to signal variance $s_k^2$. The power transfer function $p_n$ of the neural filter corresponds here to the filter power $|L_k|^2$. Finally, the negative Lagrange multiplier $-\lambda$ corresponds here to the positive Lagrange multiplier $+\Lambda$. With these correspondences, the optimal filter for channel $k$ has a gain $|L_k|$ given by:

$$|L_k|^2 = \frac{-\left(2 + s_k^2\right) + \sqrt{s_k^4 + 4 s_k^2 / \Lambda}}{2\left(1 + s_k^2\right)} \tag{0.7}$$

provided that the above quantity is non-negative, and has a gain of zero otherwise. The range of values of $s_k$ for which the above quantity is $\leq 0$ corresponds to signals that are not worthwhile to code, because the signal-to-noise is too small given the constraint on the channel dynamic range. More specifically, the above quantity is positive (and hence $|L_k|$ is nonzero) provided that $s_k > \sqrt{\Lambda / (1 - \Lambda)}$. Note that this critical value becomes infinite as $\Lambda$ approaches one from below, indicating that $\Lambda$ near one is the transmission-limited regime. Conversely, the critical value of $s_k$ approaches zero as $\Lambda$ approaches zero from above, indicating that this is the sampling-limited regime. We further discuss these regimes below.

## Transmission-limited regime

As mentioned, the transmission-limited regime corresponds to the limit of $\Lambda \to 1$ from below. For signals below the critical level of $\sqrt{\Lambda/(1-\Lambda)}$, the optimal gain is zero, and signals are not encoded. For signals that are large compared to this cutoff, the main limitation is output power. In this regime, the optimal gain is inversely proportional to the signal strength (*Figure 4—figure supplement 2A*), as the asymptotic behavior of *Equation 0.7* in the limit of large signal strength $s_k$ is:

$$|L_k|^2 \sim \frac{1/\Lambda - 1}{1 + s_k^2} \tag{0.8}$$

This is the classic 'whitening' regime, namely small signals are enhanced so that output power is equalized across channels: $|L_k| \sim 1/s_k$ for large $s_k$.

Note that when $\Lambda$ is close to 1, there is an abrupt transition between signals that are encoded in inverse proportion to their size, and signals that are too small to be encoded at all (*Figure 4—figure supplement 2A*).

## Sampling-limited regime

When $\Lambda \to 0$ from above, the transition between signals that are not encoded at all, and signals that are encoded in inverse proportion to their size, undergoes a broadening. This results in a regime in which the optimal gain *increases* with signal strength (*Figure 4—figure supplement 2B*). This regime covers signals that are only modestly above the critical level of $\sqrt{\Lambda/(1-\Lambda)}$, that is signals for which sampling noise (rather than output capacity) is the dominant constraint. The extent of this regime increases as the relative importance of the output constraint $\Lambda$ decreases toward 0.

We determine the limiting dependence of $|L_k|$ on $s_k$ from the asymptotic behavior of *Equation 0.7* in the limit of small $\Lambda$:

$$|L_k|^2 \sim \frac{s_k}{1 + s_k^2} \sqrt{\frac{1}{\Lambda}} \tag{0.9}$$

For signals that are small compared to the sampling noise ($\sqrt{\Lambda} < s_k < 1$), the optimal filter is proportional to the square root of the signal strength, $|L_k| \sim s_k^{1/2} \Lambda^{-1/4}$.

## Correspondence with perceptual sensitivity to local image statistics

We interpret the gain $|L_k|$ as representing the amount of resources devoted to a given signal $s_k$. Since it is a direct measure of signal-to-noise for a unit-size input, it therefore corresponds to perceptual sensitivity.

In the psychophysical experiments here, we measure sensitivity for each of the image statistic coordinates $\{\beta_|, \beta_-, \beta_/, \beta_\backslash, \theta_\ulcorner, \theta_\urcorner, \theta_\lrcorner, \theta_\llcorner, \alpha\}$, using a highly artificial set of stimuli. As predicted from the sampling-limited regime, we find that gains $|L_k|$ are larger for the channels in which the natural environment provides larger values of the signal $s_k$.

While this analysis provides a rigorous identification of a regime in which gain increases with signal strength, we caution that it is an asymptotic analysis of a simplified model of feature coding. It therefore stops short of making the quantitative prediction that gain (sensitivity) is proportional to the square root of the signal strength of each image statistic.

On the other hand, the analysis does translate into a quantitative prediction about perceptual axes (i.e., about the orientations of the isodiscrimination contours). As shown in *Figure 3* (blue contours), the image statistic coordinates $\{\beta_|, \beta_-, \beta_/, \beta_\backslash, \theta_\ulcorner, \theta_\urcorner, \theta_\lrcorner, \theta_\llcorner, \alpha\}$ have substantial covariances. A rotation of the coordinates will thus yield a new set of coordinates with zero covariance and independent sampling errors. If these new coordinates are independently coded, then the perceptual axes will share the same axes as the image statistics which is what we find (*Figure 3B*).

## Numerical optimization in two dimensions

Here, we numerically show that in the 2-dimensional case, the axes of the optimal encoder will be aligned with the principal axes of the input statistics. As shown in *Figure 4—figure supplement 1*, the response **r** is given by:

$$\mathbf{r} = \mathbf{L}(\mathbf{s} + \xi) + \eta, \tag{0.10}$$

where $\xi$ is the sampling noise, $\eta$ is the intrinsic channel noise, and $\mathbf{s}$ is a $d$-dimensional signal from natural scenes (each dimension corresponds to one of our image statistic coordinates; for simplicity, let $d = 2$, that is, we examine one pairwise plane). $\mathbf{L}$ is the linear transformation that we are looking for: this is essentially a 'gain' plus 'rotation' transformation. The axes of perceptual isodiscrimination contours should then be given by the eigenvalues of $\mathbf{LL}^T$. The covariance of the stimuli is $\mathbf{S} = \langle \mathbf{ss}^T \rangle$. Noise is assumed IID, given by $\langle \xi\xi^T \rangle = \Xi\mathbf{I}$ at the input and $\langle \eta\eta^T \rangle = \Sigma\mathbf{I}$ at the output, where $\mathbf{I}$ is a $2 \times 2$ identity matrix and $\Xi$ and $\Sigma$ are noise magnitudes. With this notation, the total noise covariance matrix of the output is given by:

$$\mathbf{N} = \Sigma\mathbf{I} + \Xi\mathbf{LL}^T. \tag{0.11}$$

The total variance at the output is:

$$\mathbf{r}^2 = d\Sigma + \Xi\,\mathrm{Tr}\,\mathbf{LL}^T + \mathrm{Tr}\,\mathbf{LSL}^T. \tag{0.12}$$

By analogy to the van Hateren derivation, we fix the output power. Without loss of generality, we choose its value to be unity, which sets the unit for all power measures in the system. The information for a Gaussian multivariate channel in a standard form, $\mathbf{r} = \mathbf{L}'\mathbf{s} + \eta$, is:

$$I = \frac{1}{2}\log\det\left(\mathbf{I} + \mathbf{S}^{\frac{1}{2}}\mathbf{L}'^T\mathbf{L}'\mathbf{S}^{\frac{1}{2}}\right), \tag{0.13}$$

but this is only valid when the noise $\eta$ is IID unit variance. In the present study, this is not the case: first, the noise, $\mathbf{N}$, is correlated in the two channels, because the sampling noise is mixed by $\mathbf{L}$; second, the variances are not the same in the two channels. We can, however, make a change of variables, $\mathbf{r}' = \mathbf{Or}$, such that the noise for the new output $\mathbf{r}'$ is IID unit variance. To do this, we decompose $\mathbf{N} = \mathbf{VDV}^T$ into its eigensystem, make $\mathbf{O} = \mathbf{D}^{-\frac{1}{2}}\mathbf{V}^T$, and identify $\mathbf{L}' = \mathbf{OL} = \mathbf{D}^{-\frac{1}{2}}\mathbf{V}^T\mathbf{L}$, so that we can use the standard result given in *Equation (0.13)*. The optimal linear filter is given by:

$$\mathbf{L}^* = \operatorname*{argmax}_{\mathbf{L},\mathbf{r}^2=1} \frac{1}{2}\log\det\left(\mathbf{I} + \mathbf{S}^{\frac{1}{2}}\mathbf{L}'^T\mathbf{L}'\mathbf{S}^{\frac{1}{2}}\right) \tag{0.14}$$

Since the output power is limited to 1 and channel noise $\Sigma$ feeds directly into the output power, there is no solution for $\mathbf{L}$ for $\Sigma > 0.5$ (since $d = 2$ and $\Sigma$ is the noise in each of the channels, the total output power is taken up by channel noise at $\Sigma = 0.5$). The magnitude of the sampling noise can be unbounded, since one can always select the gain in $\mathbf{L}$ to be low enough so that the constraint on total output power is satisfied. Because the gain rescales the input, we can fix the total power of the input signal (the trace of $\mathbf{S}$) to be unity. With this choice, the remaining parameters of the problem are the magnitude of the channel noise ($\Sigma$) and the magnitude of the sampling noise relative to the input power (i.e. 1/SNR at the input).

Given these two parameters that determine the sampling and channel noise magnitudes, we generate input signal covariances $\mathbf{S}$ with total power of unity but with randomly selected 'tilts' (angles of the leading eigenvector of $\mathbf{S}$ measured relative to the horizontal) and 'eccentricities' ($= \sqrt{1 - g_{min}^2/g_{max}^2}$, where $g$ are the eigenvalues of $\mathbf{S}$); these quantities can be directly estimated from natural scenes. We then use constrained optimization to numerically identify the optimal transformation $\mathbf{L}^*$. For each such solution for $\mathbf{L}^*$, we compute the eigensystem of $\mathbf{L}^*\mathbf{L}^{*T}$, extract its eccentricity and tilt as describe above, and compare these values to the eccentricity and tilt of the input signal.

We identify the following efficient coding regimes that depend the total noise and on the relative magnitudes of sampling and channel noises (*Figure 4—figure supplement 3*):

## Transmission-limited regime (total noise <0.5)

$0 \leq \Xi \ll \Sigma$ (dominating channel noise). The optimal strategy is decorrelation by whitening (*Figure 4—figure supplement 4A*); the tilt of the filter relative to the signal is $\pi/2$, and the eccentricities are equal (i.e., the small eigenvalue of $\mathbf{L}^*\mathbf{L}^{*T}$ is proportional to the inverse of the large eigenvalue of $\mathbf{S}$ and vice versa, indicating that the gain scales as the inverse of the input power).

$0 < \Xi \ll 1, \Sigma = 0$ (zero channel noise, small sampling noise). The optimal strategy is still decorrelation (*Figure 4—figure supplement 4B*) with signal components of higher power being suppressed by the gain, but the suppression does not follow the inverse law as above.

## Sampling-limited regime (total noise >0.5)

$\Xi \geq 1, \Sigma = 0$ (zero channel noise, large sampling noise). The tilt of the filter matches the tilt of the signal, and the gain scales with input power. For high sampling noise and zero channel noise, the gain scales as the square-root of the input power (*Figure 4—figure supplement 4C*).

$\Xi > \Sigma > 0$ (dominating sampling noise). In a broad regime of noise strengths where sampling noise dominates over non-zero channel noise, the tilt of the gain matches the tilt of the signal, and the gain roughly scales with the input power (*Figure 4—figure supplement 4D*). This regime is consistent with the correspondence that we observe between the natural scenes statistics and the psychophysical measurements.

## Acknowledgements

We thank Jason Prentice and Eizaburo Doi for valuable discussions. This work was supported by NIH EY07977, NSF PHY-1058202, FWF P25651, NEI Vision Training Grant 5-T32-EY007035-32, and the Fondation Pierre Gilles de Gennes.

## Additional information

### Funding

| Funder | Grant reference number | Author |
|---|---|---|
| National Eye Institute | EY07977 | Jonathan D Victor |
| National Science Foundation | PHY-1058202 | Vijay Balasubramanian |
| Austrian Science Fund | FWF P25651 | Vijay Balasubramanian, Gašper Tkačik |
| National Eye Institute | Vision Training Grant 5-T32-EY007035-32 | Vijay Balasubramanian |
| Fondation Pierre Gilles de Gennes | | Vijay Balasubramanian |

The funders had no role in study design, data collection and interpretation, or the decision to submit the work for publication.

### Author contributions

AMH, JJB, MMC, JDV, VB, GT, Conception and design, Acquisition of data, Analysis and interpretation of data, Drafting or revising the article, Contributed unpublished essential data or reagents

### Ethics

Human subjects: The human subjects research (visual psychophysics) was approved by the Institutional Review Board of the Weill Cornell Medical College, and was in accord with the World Medical Association Declaration of Helsinki. Informed consent was obtained from each subject prior to the experimental sessions, and consent to publish was obtained from Mary Conte (MC), the one subject who is potentially identifiable by the initials since she is also an author.

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
