## [Decision Letter]

Thank you for sending your work entitled “Variance is salience: efficient coding in central sensory processing” for consideration at *eLife*. Your article has been evaluated by Eve Marder (Senior editor), Timothy Behrens (Reviewing editor), and 3 reviewers.

The Reviewing editor and the other reviewers discussed their comments before we reached this our decision, and the Reviewing editor has assembled the following comments to help you prepare a revision.

All reviewers found the data interesting, but all had concerns about whether they addressed the central hypotheses outlined in the abstract and introduction. In brief we do not find a convincing argument that the results 1) reflect a central mechanism, 2) relate to behavior, or 3) make different predictions than what one would predict for the sensory periphery given the efficient coding hypothesis. These questions are specified in detail in the reviews below. The consensus that emerged from the consultation session is that clarification and rewriting will be needed to ensure that the contribution of this work can be understood. Ordinarily *eLife* does not provide the full reviews, but in this case we believe it is important that you see the concerns of the reviewers as you prepare your revision.

Reviewer #1

The authors show that psychophysical detection thresholds for figure/ground segmentation of image patches in noise are extremely well predicted by the extent to which the features defining the patch are variable in natural images. Both follow a characteristic pattern in which cardinal, then oblique features are the most variable over images (and the most detectable), followed by fourth order statistics, followed by third order (L shapes in a 4-pixel glider). The correspondence between lines of isosensitivity and the precision matrix for pairwise comparisons in 10 dimensions are very striking. The authors interpret their findings as showing that efficient coding - heightened sensitivity to the most informative features of the world - occurs not only at a sensory level but at a central decision level.

I liked the paper and approach very much. I was a little more doubtful about the novelty of the findings. My concerns are as follows:

1) I wasn't sure how the paper demonstrated that efficient coding operates at a 'central' level, as the authors claim in the abstract In fact, the paper doesn't really have anything to do with what most cognitive scientists think of as 'central processing'. It shows that we are most sensitive to patches defined by cardinal orientations, and that variability along this axis is an informative way to characterise natural scenes.

2) The psychophysical results seems in part to restate the well-known oblique effect. It is known that we are most sensitive to cardinal orientations, and the neural underpinnings of this effect (in visual cortex) have been much discussed. The work of Simoncelli and colleagues (cited appropriately here) has been key in pointing out that sensitivity to cardinal directions matches the prevalence of cardinal information in the world. I think the work here extends that finding by demonstrating a particularly good correspondence between the spatial correlations in image statistics and this sensitivity over a number of basic features.

In short, I think this is an interestingly, well-presented paper that I would encourage others to read, and that I would cite in my own work. However, I am not sure whether it is telling us something we didn't know, or just demonstrating something we did know in a particularly elegant fashion.

Reviewer #2

This paper describes a robust and interesting observation that links properties of natural images to psychophysical properties of the visual system. While this observation is certainly worthwhile publishing, I'm less convinced about the author's conclusion with regard to “variance is salience” as an efficient coding principle for central sensory processing.

While the authors correctly note that central processing must be concerned with behaviorally relevant aspects of the inputs, the 'variance' in their principle is completely independent of the organism's behavior and purely defined in terms of the stimulus. In fact, none of their discussion nor references to the earlier van Hateren papers that the authors invoke as a framework for interpreting their findings involves behavior. I suggest to simply drop that part of their claim and the suggestion that their principle might account for central processing and describe it as a principle governing early sensory processing where different constraints apply than in the sensory periphery. (Unless I'm missing a central part of their argument, of course. Example below.)

I think the paper would benefit greatly from a more explicit setup, ideally including a Figure, of the various potential constraints faced by early sensory processing and their respective implications for their experiments. While this aspect is not the main contribution of their paper, it would make it more explicit which alternative models/constraints are being excluded by their observation. It would also delineate better, what is inevitable conclusion within the efficient coding framework (finite resources and high input noise), and what is interpretation/speculation. Both are currently mixed in the Discussion section.

Psychophysical measurements necessarily characterize the entire system; from periphery to behavior. The author suggest their results support an optimization principle separate and different from that in the periphery and while I agree on an intuitive level, I would welcome help in thinking through the implications of combining two different principles, applied after each other, for the aggregate quantities that are being measured. E.g. why does the high input noise constraint only refer to what enters cortex, not already the retina, overriding the output-bottleneck in the visual periphery?

Finally, I'd like to commend the authors on making excellent use of the possibility of including supplementary information.

In summary, I recommend publication of this paper subject to:

1) Providing a more explicit setup for the different hypotheses to be tested and how they derive from the efficient coding principle subject to different constraints.

2) Omitting the behaviorally relevant claims not supported by their framework.

3) Responses to more minor points of critique below.

Reviewer #3

The authors demonstrate that the level of variability in various local, multi-point statistics in natural scenes quantitatively predicts the relative salience of these statistics. I find their data largely convincing, and I feel that this is an important question and that the authors are careful and creative in their approach here. However, I have several concerns, which I suspect in part reflect my own ignorance or misunderstanding about the present paper and past work by these authors.

First, I am not clear on what is qualitatively new here beyond what was shown in the nice 2010 PNAS paper by several of the same authors. I can see that some of the details of their analyses are different, and the comparisons here are more quantitative, but it seems to me that the main ideas and conclusions were already in place in the previous work.

The authors pitch their results in terms of cortical processing, but the only data presented are from human psychophysics, used here as a proxy for cortical processing. The authors give arguments for why salience strongly reflects cortical processing. But I feel that the claims about cortical processing are stated too strongly in some parts of the paper given that they do not directly measure cortical activity nor perform any manipulations to cortex, for example.

It seems to me that the dichotomy set up by the authors between predictions for peripheral and central processing is somewhat overstated. For example, in the final sentence of the Abstract, they state that the efficient coding hypothesis applies in “a different guise” and makes “different predictions” for the central nervous system than the periphery. They predict that sensitivity in the CNS should be greater for highly variable stimuli, but it is not clear to me that this is not one of the predictions one would make for peripheral sensory neurons using this principle.

What was the exact task being performed by the subjects?

What would Figure 1 look like for IId samples from white (or colored) noise?

How do the results change for images pixelated along a grid that is 45° tilted relative to the vertical and horizontal directions?

Related to this, could the √2 differences in the degree of variation for the vertical/horizontal and diagonal beta data points in Figure 3 be due to the difference in the distance between the points in question given the orientation of the pixelation grid?

---

## [Author Response]

*All reviewers found the data interesting, but all had concerns about whether they addressed the central hypotheses outlined in the abstract and introduction. In brief we do not find a convincing argument that the results 1) reflect a central mechanism, 2) relate to behavior, or 3) make different predictions than what one would predict for the sensory periphery given the efficient coding hypothesis. These questions are specified in detail in the reviews below. The consensus that emerged from the consultation session is that clarification and rewriting will be needed to ensure that the contribution of this work* can *be understood. Ordinarily* eLife *does not provide the full reviews, but in this case we believe it is important that you see the concerns of the reviewers as you prepare your revision*.

Response to broad concerns:

1) To what extent do our results reflect a central mechanism?

There are three lines of evidence, and we detail these below and in the manuscript. First, stimuli are all high contrast (100%) and of a readily visible size (14 arcmin), so retinal limitations of contrast sensitivity and resolution are eliminated. Second, the task requires pooling of information over wide areas. Third, extracting three- and four-point correlations requires a kind of nonlinear processing (two stages of nonlinearity) that is not generally considered to be present in the retina; physiologic recordings show that neural responses to these correlations are not present in the thalamus, but are present in visual cortex, and much more so in V2 than in V1. We acknowledge that these lines are indirect; we do not directly record cortical activity, and we now state this explicitly in the manuscript (see response to comments raised by Reviewer 3). But we believe that together, these lines of evidence justify the statement that the thresholds are determined by limitations of central processing.

2) To what extent do our results relate to behavior?

The goal of the visual system is to make discriminations that are useful to guide action, rather than to reconstruct the image per se. We had used the term “behaviorally relevant” to highlight this idea, once in the Abstract and once in the manuscript; we have now removed it from the Abstract, where we see how it might have been confusing, and we expand on the point in the Introduction.

With regard to the comments raised by Reviewer 2, obviously our analysis of natural image statistics was without regard to task, and the psychophysical task was not designed to mimic a real-life behavior. But we think this is a strong point, not a weakness, as it shows that the allocation of resources in early cortical visual processing can be accounted for in a general framework. We now comment on this in the text.

3) To what extent do our results make different predictions than one would predict for the sensory periphery?

We now clarify this at several points in the manuscript, and we add a new Figure. Briefly, the contrast is not between peripheral and central processing per se, but the difference between two efficient coding regimes: one that is limited by noise in the output (i.e., limited capacity) and one regime that is limited by noise in the input (i.e., limited sampling). The periphery is typically considered to be characterized by the first regime, at least when it comes to thinking about receptive field properties, and we hypothesize that central processing is characterized by the second. The new Figure illustrates the qualitative difference between these regimes: when output noise is limiting, efficient coding predicts a whitening (reduced sensitivity for more variable input components); when input noise is limiting, efficient coding predicts the opposite (increased sensitivity for more variable input components).

Since central processing follows the peripheral processing, one could wonder how the two stages of the same processing stream can be subject to different constraints. The reason is as follows: the “input signal” in the context of efficient coding for the retina is directly the raw light intensity, and efficient coding regime is derived from the spatiotemporal correlation structure of light intensity. In contrast, the input signals for central coding are nonlinear functions of the original image (image statistic coordinates); this means that efficient coding regime is derived from the covariance structure of image statistics. Since the formal inputs to the encoders are very different (light or contrast in the periphery, image statistic estimates centrally), a different regime of efficient coding emerges.

Reviewer #1

*[…] In short, I think this is an interestingly, well-presented paper that I would encourage others to read, and that I would cite in my own work. However, I am not sure whether it is telling us something we didn't know, or just demonstrating something we did know in a particularly elegant fashion*.

We are very pleased that Reviewer 1 found our paper to be both interesting and well-presented. We do feel that the paper is telling us something new and rather exciting, as we explain below in responses to the specific comments.

1) I wasn’t sure how the paper demonstrated that efficient coding operates at a ‘central’ level, as the authors claim in the abstract. In fact, the paper doesn’t really have anything to do with what most cognitive scientists think of as ‘central processing’.

With regard to ‘central processing’: We use this term to distinguish our focus – cortical sensory processing – from processing in the sensory periphery. We think this is a reasonable use of the term, even though there is (of course) much additional central processing that we do not consider. But we understand the potential for confusion, and we now clarify the use of the term in the Introduction: “To test this hypothesis, we focus on early stages of central visual processing. Here, early visual cortex (V1 and V2) is charged with extracting edges, shapes, and other complex correlations of light between multiple points in space...”

We feel that there is strong evidence that the computations that determine task performance occur in primary visual cortex, both V1 and V2, and we expand on this point in the paper. Specifically, we now state:

Discussion: “Although we did not record cortical responses directly, several lines of evidence indicate that that the perceptual thresholds we measured are determined by cortical processes. […] Conversely, macaque visual cortical neurons (Purpura et al., 1994), especially those in V2, manifest responses to three- and four-point correlations (Yu et al., 2013). ”

*It shows that we are most sensitive to patches defined by cardinal orientations, and that variability along this axis is an informative way to characterise natural scenes*.

Our results about sensitivities to pairwise statistics in horizontal, vertical, and diagonal directions go substantially beyond a demonstration of previously-known findings, in several ways.

First, the “on-axis” sensitivities are not obvious consequences of the orientation tuning of cortical neurons (since we consider both positive and negative correlations) or the oblique effect (our observed differences are substantially larger, likely because these configurations differ in more than orientation). For further discussion of the oblique effect, please see our response to point 2 below.

Second, only a small portion of our findings (a prediction of four relative sensitivities) relate to individual “axes.” Most of our findings (more than a dozen free parameters) relate to how the axes interact – again, not something that is a consequence of orientation-tuning or the oblique effect.

Finally, it is possible that there is a misunderstanding about the axes themselves: they are not the spatial axes within an image, but abstract axes within a space of image statistics. We can see how this might have been confusing, and we now clarify it with the following text:

Results: “When pairs of natural image statistics covary, thus sampling oblique directions not aligned with the coordinate axes in the space of image statistics, our hypothesis predicts that human perceptual sensitivity is matched to both the degree and the direction of that covariation (we are referring here to the orientation of a distribution in the coordinate plane of a pair of image statistics, and not to an orientation in physical space).”

An important way in which our results extend beyond what is already known relates to the predictions of normative theories. To date, normative theories applied to cortical processing have been focused on predicting the response properties of single cells, considered as linear or quasilinear filters. Here, we ask how resources should be distributed across a population of cells to represent a diverse set of nonlinear, higher-order features that cover a multidimensional domain. We highlight this goal within the Introduction:

Introduction: “Can we extend such theories beyond the sensory periphery to describe cortical sensitivity to complex sensory features? […] We will show that these ideas predict a specific organizing principle for aggregate sensitivities arising in cortex: the perceptual salience of complex sensory signals increases with the variability, or unpredictability, of the corresponding signals over the ensemble of natural stimuli.”

We additionally emphasize that the goal of our study is to measure how sensitivities are distributed across a set of features, and is not limited to identifying features for which sensitivity is greatest. We now highlight this within the Introduction and Discussion:

Introduction: “We compare the spatial variation of local patterns of light across natural images with human sensitivity to manipulations of the same patterns in synthetic images. This allows us to determine how sensitivity is distributed across many different features, rather than simply determining the most salient ones. To this end...”

Discussion: “How should neural mechanisms be distributed to represent a diverse set of informative sensory features? We argued that...”

*2) The psychophysical results seems in part to restate the well-known oblique effect. It is known that we are most sensitive to cardinal orientations, and the neural underpinnings of this effect (in visual cortex) have been much discussed. The work of Simoncelli and colleagues (cited appropriately here) has been key in pointing out that sensitivity to cardinal directions matches the prevalence of cardinal information in the world. I think the work here extends that finding by demonstrating a particularly good correspondence between the spatial correlations in image statistics and this sensitivity over a number of basic features*.

While we can see how our findings might at first glance be considered to be part of an “oblique effect,” they are actually quite different. The effect size we report is much larger: sensitivities are about 50% higher for cardinal directions vs diagonal ones; in the classical oblique effect, sensitivity for oblique- vs cardinal-direction gratings differ by 10–20% in the midrange of spatial frequencies (Campbell et al., 1966). The reason that the effect size we observe is much larger is that the horizontal and vertical pairwise correlations differ from the diagonal pairwise correlations in more ways than just orientation: checks involved in horizontal and vertical pairwise correlations share an edge, while checks involved in diagonal pairwise correlations only share a corner. Furthermore, the image property relevant to explaining the oblique effect in the work of Simoncelli and colleagues are the local statistics of image gradients / oriented edges. While pairwise correlations between pixels (that we study) can influence image gradients (responsible for explaining the oblique effect) and vice versa, there is no simple one-to-one mapping between the two sets of statistics (e.g., high beta-vertical does not by itself imply a lot of vertical edge fragments). We now explain how our findings differ from the oblique effect:

Results: “Note that the difference between the sensitivities in the horizontal and vertical directions (β− and β|) vs the diagonal directions (β\ and β/) is not simply an “oblique effect”, i.e., a greater sensitivity for cardinally- vs obliquely-oriented contours (Campbell et al., 1966). Horizontal and vertical pairwise correlations differ from the diagonal pairwise configurations in more than just orientation: checks involved in horizontal and vertical pairwise correlations share an edge, while checks involved in diagonal pairwise correlations only share a corner. Correspondingly, the difference in sensitivities for horizontal and vertical correlations vs diagonal correlations is approximately 50%, which is much larger than the size of the classical oblique effect (10–20%) (Campbell et al., 1966).”

Additionally, much of our study is devoted to third and fourth-order correlations and the interactions of the texture statistics. These statistics are qualitatively different from any pairwise correlations that might be relevant for the oblique effect, and are crucial for our demonstration that efficient coding also applies beyond the pairwise order.

In sum, we agree that the main finding of the work can be framed in the context of the match between image statistics and visual sensitivities (as in the Simoncelli framework), but we think that it is important to emphasize that we are not looking at an “oblique effect.”

Reviewer #2

[…] In summary, I recommend publication of this paper subject to:

*1) Providing a more explicit setup for the different hypotheses to be tested and how they derive from the efficient coding principle subject to different constraints*.

*2) Omitting the behaviorally relevant claims not supported by their framework*.

*3) Responses to more minor points of critique below*.

We are very pleased that Reviewer 2 recommends publication of our paper subject to his/her outlined changes. Below, we address each of his/her concerns: (**A**) is addressed in points 1-2 below, (**B**) is addressed in points 3 and 9 below, and (**C**) is addressed in points 4-8 below.

*1) I think the paper would benefit greatly from a more explicit setup, ideally including a figure, of the various potential constraints faced by early sensory processing and their respective implications for their experiments. While this aspect is not the main contribution of their paper, it would make it more explicit which alternative models/constraints are being excluded by their observation. It would also delineate better, what is inevitable conclusion within the efficient coding framework (finite resources and high input noise), and what is interpretation/speculation. Both are currently mixed in the Discussion section*.

We agree. We added substantial clarifying material and a new figure (Figure 4), as described below.

2) Psychophysical measurements necessarily characterize the entire system - from periphery to behavior. The authors suggest their results support an optimization principle separate and different from that in the periphery and while I agree on an intuitive level, I would welcome help in thinking through the implications of combining two different principles, applied after each other, for the aggregate quantities that are being measured. E.g. why does the high input noise constraint only refer to what enters cortex, not already the retina, overriding the output-bottleneck in the visual periphery?

If we understand the reviewer’s comment, it can be rephrased as follows: how can the cortex be operating in a regime dominated by input noise, when it follows peripheral processing, which sees the same input but is dominated by output noise?

The resolution of this apparent paradox is twofold. First, the critical issue is not the amount of input noise, but rather its size relative to the output noise. So, even though there is input noise at the level of the visual input, its severe output restriction (the transmission bottleneck) means that peripheral processing may be operating in a regime dominated by output noise. For cortex, which does not have the same output bottleneck, the same amount of input noise can dominate. The second consideration is the following. As we and others suggest, we view the “job” of early cortical processing as making inferences about surfaces and materials from visual texture (an idea that we formalize by measuring sensitivity to local image statistics). Given this goal, the fact that cortex only has access to a limited sample of a texture (i.e., the sample presented by a particular object) is also a form of input noise.

To illustrate this point, consider the following “toy visual system” whose task is to detect and estimate the local coherent motion of small, randomly moving point-like objects that are of high contrast (for example, the classic task of Newsome and Pare (1988), but with the dots placed on a 1/f background). Neurons in the periphery efficiently code the dots, making use of decorrelation. Centrally, however, the object of interest is the correlation structure of moving dots in small image patches, to extract the local coherent motion component. For this task, central processes must compare counts of how many dots move in each direction; at low density of dots, these estimates will be subject to large sampling errors due to random arrival and departure of dots from the region that is being sampled. Importantly, no matter what the SNR of the input is (even in 100% contrast vision where each dot is perfectly resolved), the local coherence estimates will be noisy due to the fact that local dot detections come as rare (and random) events; and the efficient central processing will thus be subject to large input noise. In our case, the local image statistics play the role of local dot counts, but the basic idea, that sampling is a form of noise, is identical. In both cases, because the signals relevant to periphery and cortex are different, one can end up in a different efficient coding regime at each stage of visual processing.

As suggested by the reviewer, we have added a new figure (Figure 4) to help clarify these points. Figure 4 illustrates the emergence of two efficient coding regimes in a system that operates under a total power (resource) constraint. The relevant regime, which depends on the relative strengths of input and output noise, reflects the constraints faced by the system in consideration. For our system (modeled as in Figure 4), we take inputs to be nonlinear image features that have already undergone peripheral preprocessing, as caricatured by our image analysis pipeline. We then apply the efficient coding framework to this set of nonlinear features, and we ask how sensitivities should be distributed among them.

The key point is that because these image features (counts of glider colorings) must be sampled from a spatial region of an image, any estimation of these features will be limited by sampling noise. We hypothesize that this sampling noise, which is a form of input noise, dominates. This predicts (as shown in Figure 4), that sensitivity should increase with signal variability. We test this prediction and show that it holds.

We reorganized and added to the text in order to illustrate this:

Discussion: “A simple model of efficient coding by neural populations is shown in Figure 4 (details in Methods, Two regimes of efficient coding). Here, to enable analytical calculations, we used linear filters of variable gain and subject to Gaussian noise to model a population of neural channels encoding different features. […] The relative sizes of input and output noise (controlled by Λ in Figure 4) determines the input ranges over which the two qualitatively different regimes of efficient coding apply.”

In addition, we clarified our language throughout the Discussion (see Discussion, Cortex faces a different class of challenges than the periphery ) to emphasize that the efficient coding regimes are a general feature of information optimization, and neither regime is a priori restricted to peripheral vs central processing. Thus, even in sensory periphery, there exist specific scenarios in which the signal is input- noise dominated. The best known example is perhaps that of night vision, when photon shot noise is no longer negligible and qualitatively reshapes peripheral information processing. Instead of decorrelation by center-surround receptive fields, receptive fields average the signal, i.e., they apply gain that enhances the correlations already present in the input in order to fight the detrimental effects of input noise.

We would like to emphasize clearly that the constraints considered here (nonlinear image features subject to sampling noise) thus differ from those considered by some of the most successful applications of efficient coding in the periphery. Because of these constraints, our finding that sensitivity increases with signal variability differs qualitatively from the findings of these studies, where whitening is optimal (or near optimal, see Doi and Lewicki (2014)) and sensitivity decreases with signal variability. Moreover, since feature extraction is thought to be one of the main tasks of central vision, the sampling noise constraint is likely to represent a general aspect of cortical processing, rather than a special case as for night vision in the periphery. And finally, the predictions made are quite different; we predict sensitivity to a wide variety of nonlinear features and their combinations, not the effective filtering behavior (i.e., shape) of a quasilinear receptive field.

*3) While the authors correctly note that central processing must be concerned with behaviorally relevant aspects of the inputs, the ‘variance’ in their principle is completely independent of the organism’s behavior and purely defined in terms of the stimulus. In fact, none of their discussion nor references to the earlier van Hateren papers that the authors invoke as a framework for interpreting their findings involves behavior. I suggest to simply drop that part of their claim and the suggestion that their principle might account for central processing and describe it as a principle governing early sensory processing where different constraints apply than in the sensory periphery. (Unless I’m missing a central part of their argument, of course. Example below*.*)*

We don’t intend to suggest that we are studying behavior, and we have edited the manuscript so that this is completely clear. We also added material to the Introduction and Discussion, including a new figure, that frames the work as consequences of differing constraints on peripheral and central processing.

Even though our focus is on visual processing and perception, the fact that visual processing must ultimately guide action plays a fundamental role: it motivates the idea that the visual system needs to make useful distinctions about the outside world, rather than reconstruct the image. But we can see that the term “behaviorally relevant” might have been misleading, so we have removed the term from the abstract, and we have explained the idea more carefully in the introduction:

“Can we extend such theories beyond the sensory periphery to describe cortical sensitivity to complex sensory features? Normative theories have been successful in predicting the response properties of single cells, including receptive fields in V1 (Olshausen and Field, 1996; Bell and Sejnowski, 1997; van Hateren and van der Schaaf, 1998; van Hateren and Ruderman, 1998; Hyvarinen and Hoyer, 2000; Vinje and Gallant, 2000; Karklin and Lewicki, 2009) and spectro-temporal receptive fields in primary auditory cortex (Carlson and DeWeese, 2002, 2012). […] We will show that these ideas predict a specific organizing principle for aggregate sensitivities arising in cortex: the perceptual salience of complex sensory signals increases with the variability, or unpredictability, of the corresponding signals over the ensemble of natural stimuli.”

Also (as mentioned in the response to the editors), we think that the fact that the image analysis and psychophysics was without regard to task is a strong point, not a weakness. We find it quite remarkable that we see such a striking match between natural image statistics and perceptual sensitivity without including any notion of “value” as related to behavior. This finding, that higher order percepts emerge from efficient coding without explicit specification of the goal, was the original hope of the efficient coding framework as first put forward by Barlow and Attneave (Barlow, 1959, 1961; Attneave, 1954). It remains an interesting direction for future research to ask how the inclusion of value would shape the framework studied here. We now mention this explicitly in the Discussion:

“Finally, we emphasize that although we have focused on perception of image statistics, we do this with the premise that this process is in the service of inferring the materials and objects that created an image and ultimately, guiding action. Thus, it is notable that we found a tight correspondence between visual perception and natural scene statistics without considering a specific task or behavioral set, and indeed, the emergence of higher order percepts without explicit specification of a task was the original hope of the efficient coding framework as first put forward by Barlow and Attneave (Barlow, 1959, 1961; Attneave, 1954). Doubtless, these “top-down” factors influence the visual computations that underlie perception, and the nature and site of this influence are an important focus of future research.”

*4) In my understanding, the black/white asymmetry is the main systematic deviation from the prediction that was empirically found. I think this should be included (with Figure) and discussed in the main manuscript and not just in the SI*.

We feel that the reviewer might have misunderstood our results in this regard: the main systematic deviation from prediction concerns positive and negative values of the fourth-order statistic α, not black/white asymmetry (the latter was eliminated by our preprocessing, in which we binarize image patches at the pixel intensity median such that each patch has equal numbers of black and white pixels (see Results, Analyzing local image statistics in natural scenes).

Specifically, there is an asymmetry of the distribution of α-values extracted from natural images that is not mirrored in psychophysical sensitivities. We mention this in Discussion, Clues to neural mechanisms, where we reference the relevant figure (Figure 3—figure supplement 8). We now also include the discussion of this asymmetry within the main manuscript, in Methods, Asymmetries in distributions of natural image statistics.

5) If your “glider” is anything more than a 2 × 2 window, then please explain in more detail.

The reviewer is correct: the glider is a 2 × 2 window. We now clarify this in two places in the main text:

Results: “As we recently showed, some informative local correlations of natural scenes are captured by the configurations of luminances seen through a “glider”, i.e., a window defined by a 2 × 2 square arrangement of pixels (Tkačik et al., 2010).”

Results: “We characterize each patch by the histogram of 16 binary colorings (2^2×2^) seen through a square 2 × 2 pixel glider.”

*6) Block average should be explained better in the main text; wouldn’t have understood it without resorting to SI*.

We now clarify the block-averaging process within the main text:

Results: “We preprocess the image patches as shown in Figure 1. This involves first averaging pixel luminances over a square region of N × N pixels, which converts an image of size L_1_ × L_2_ pixels into an image of reduced size L_1_/N × L_2_/N pixels. Images are then divided into R × R square patches of these downsampled pixels and whitened (see Methods, Image preprocessing, for further details).”

7) “Single scaling parameter for each image analysis”: what exactly does that mean? How many parameters per subject and comparison are needed? Can you say anything interesting about their distribution and the correlations between them?

We clarify these points in the main text, including a rewording of the quoted phrase. As we explain, we carried out image analyses for several choices of two size parameters: a block-average factor N (where N × N image pixels are averaged at the first step of the processing pipeline) and a patch size R (where R × R is the size of a patch in N × N pixels in which statistics are determined). There were 2 choices of N (2 and 4) and 3 choices of R (32, 48, and 64), for a total of 6 parallel image analyses (in the SI, we show that our results hold over a wider range too). For each of these analyses, there was just one scale factor (the same for all statistics and all subjects), adjusted to minimize the least-squares error between the set of standard deviations and the set of psychophysical sensitivities:

Results: “This qualitative comparison can be converted to a quantitative one (Figure 3), as a single scaling parameter aligns the standard deviation of natural image statistics with the corresponding perceptual sensitivities. In this procedure, each of the six image analyses is scaled by a single multiplicative factor that minimizes the squared error between the set of standard deviations and the set of subject-averaged sensitivities (see Methods, Image preprocessing, and Figure 3—figure supplement 1 for additional details regarding scaling).”

With regard to a relationship between the scale factors and the analysis parameters (the block-average factor N and patch size R), there is (as one might expect) a systematic relationship. We now show this in detail in Methods, Image preprocessing, with a new figure and new text:

Methods: “To compare between natural image and psychophysical analyses, we scale the set of 9 standard deviations extracted from a given image analysis by a multiplicative factor that minimizes the squared error between the set of 9 standard deviations and the set of 9 psychophysical sensitivities. Figure 3—figure supplement 1 shows the value of the scale factor for different choices of the block-average factor N and patch size R.”

Figure 3—figure supplement 1. Scaling of natural image analyses. We scale each image analysis by a single scale factor that minimizes the squared error between the set of 9 standard deviations and the set of 9 psychophysical sensitivities. The scale factors are shown here as a function of block-average factor N for different choices of the patch size R. We find that the variance of image statistics decreases with increasing values of N, and thus larger values of N require a larger scale factor. Similarly, for a given value of N, the variance of image statistics increases with increasing R, and thus larger values of R require a larger scale factor.

8) I’d welcome it if the authors could spell out the Null-hypothesis for their significance tests, rather than saying what they did mechanistically (”coordinate labels independently shuffled”).

We now do this within the main text by replacing the mechanistic description of the significance tests with a description of each null hypothesis:

Results: “This value ranges from 0.987 to 0.999 across image analyses and was consistently larger than the value measured under the null hypothesis that the apparent correspondence be- tween statistics and sensitivities is chance (p ≤ 0.0003 for each image analysis; see Methods, Permutation tests, for details regarding statistical tests).”

Results: “This value, averaged across coordinate planes, ranges from 0.953 to 0.977 across image analyses. We compared this correspondence to that obtained under the null hypotheses that either (i) the apparent correspondence between image statistic covariances and isodiscrimination contours is chance, or (ii) the apparent covariances in image statistics are due to chance. The observed correspondence is much greater than the value measured under either null hypothesis (p ≤ 0.0003 for each image analysis under both hypotheses; see Methods, Analysis of image statistics in pairwise coordinate planes, and Figure 3—figure supplement 2 for comparisons of eccentricity and tilt, and Methods, Permutation tests, for statistical tests).”

We also made corresponding additions to Methods, Permutation tests, in order to supplement the mechanistic description of the significance tests with the full descriptions of the null hypotheses.

*9) Discussion: “Cortical mechanisms should be...”: This is in clear contradiction to the behaviorally relevant argument elsewhere in the paper. Very variable features may be behaviorally relevant demanding cortical resources, or they may not be asking for them to be factored/normalized out (e.g. overall brightness). The theoretical framework is all about maximizing information with respect to the inputs, not with respect to behaviorally relevant outputs*.

We agree, this sentence did not properly explain our ideas. We rewrote this section as follows:

Discussion: “How should neural mechanisms be distributed to represent a diverse set of informative sensory features? We argued that, when performance requires inferences limited by sampling of the statistics of input features, resources should be devoted in proportion to feature variability. A basic idea here is that features that range over a wider range of possible values are less predictable, and will better distinguish between contexts in the face of input noise. We used this hypothesis to successfully predict...”

Reviewer #3

*The authors demonstrate that the level of variability in various local, multi-point statistics in natural scenes quantitatively predicts the relative salience of these statistics. I find their data largely convincing, and I feel that this is an important question and that the authors are careful and creative in their approach here. However, I have several concerns, which I suspect in part reflect my own ignorance or misunderstanding about the present paper and past work by these authors*.

*First, I am not clear on what is qualitatively new here beyond what was shown in the nice 2010 PNAS paper by several of the same authors. I* can *see that some of the details of their analyses are different, and the comparisons here are more quantitative, but it seems to me that the main ideas and conclusions were already in place in the previous work*.

Briefly, the previous work (Tkačik et al., 2010) demonstrated a close relationship between which image statistics are analyzed by the visual system, and the statistics of natural images: resources are devoted to image statistics that cannot be predicted from simpler ones (Tkačik et al., 2010). Here, we build on this, and examine allocation of resources within this identified set of informative images. The previous work made no attempt to predict this allocation, and the current work is not applicable to image statistics that are uninformative (the distinction made in the previous work).

Previous work considered natural scene statistics defined based on local correlations defined over different spatial configurations of pixels. It demonstrated that these statistics can be divided into two groups, informative and uninformative. Informative features – higher-order statistics whose value cannot be deduced from lower-order statistics – are encoded, while uninformative features are not. Indeed, the visual system would be wasteful if it were to invest in mechanisms to encode uninformative statistics, and evidence from Tkačik et al. (2010) suggests that this does not happen. This advance was crucial; as one looks at high-order statistics, there is an exponential explosion of correlations that we (or the brain) could compute. Tkačik et al. (2010) showed that it only makes sense to focus on ones that are informative, and that “informativeness” can be determined from natural scenes alone.

The current manuscript takes this finding as a starting point for exploring the relative allocation of resources to encoding the set of nine informative statistics defined by the 2 × 2 square glider. Unlike Tkačik et al. (2010), which simply identified this set of statistics as informative, here we make and confirm dozens of quantitative predictions concerning how resources are allocated to encode them. The fact that an application of the efficient coding principle is successful in characterizing resource allocation is by no means a straightforward consequence of Tkačik et al. (2010), as any allocation of resources within this parameter set would have been consistent with those previous results.

To clarify this distinction, we have schematized the existence of informative vs uninformative features in a newly-added figure (Figure 4), and in the associated text. We note that the existence of uninformative statistics was the focus of previous work, and we note that the present work focuses on the optimal allocation of resources among those statistics that are informative:

Results: “Successive stages of sensory processing share the same broad goals: invest resources in encoding stimulus features that are sufficiently informative, and suppress less-informative ones. In the periphery, this is exemplified by the well-known suppression of very low spatial frequencies; in cortex, this is exemplified by insensitivity to high-order correlations that are predictable from lower-order ones. Previous work has shown that such higher-order correlations can be separated into two groups – informative and uninformative – and only the informative ones are encoded (Tkačik et al., 2010). We used this finding to select an informative subspace for the present study, and we asked how should resources should be efficiently allocated amongst features within this informative subspace.”

*The authors pitch their results in terms of cortical processing, but the only data presented are from human psychophysics, used here as a proxy for cortical processing. The authors give arguments for why salience strongly reflects cortical processing. But I feel that the claims about cortical processing are stated too strongly in some parts of the paper given that they do not directly measure cortical activity nor perform any manipulations to cortex, for example*.

We rewrote this section, acknowledging that the inference is indirect and explicitly stating that we did not record cortical responses. But we do feel that the evidence that psychophysical performance reflects cortical processing is compelling, as it is based on independent lines of evidence that, individually, are strong: briefly, stimuli are many times above contrast and resolution thresholds; task performance requires pooling over wide areas; the computations required for the task are more complex than what is considered to occur in the retina; and physiologic recordings show that neural responses to high-order correlations are not present in the thalamus, but are present in visual cortex, and much more so in V2 than in V1.

Specifically, we state:

Results: “Although we did not record cortical responses directly, several lines of evidence indicate that that the perceptual thresholds we measured are determined by cortical processes. […] Conversely, macaque visual cortical neurons (Purpura et al., 1994), especially those in V2, manifest responses to three- and four-point correlations (Yu et al., 2013).”

Because this evidence is admittedly indirect, we added the modifier “likely” to the subheader for this section, to read “Perceptual salience of multi-point correlations likely arises in cortex.”

*It seems to me that the dichotomy set up by the authors between predictions for peripheral and central processing is somewhat overstated. For example, in the final sentence of the Abstract, they state that the efficient coding hypothesis applies in “a different guise” and makes “different predictions” for the central nervous system than the periphery. They predict that sensitivity in the CNS should be greater for highly variable stimuli, but it is not clear to me that this is not one of the predictions one would make for peripheral sensory neurons using this principle*.

We see how a ‘dichotomy’ is an oversimplification, and a fairer description is that of two qualitatively different regimes (“whitening” and input-noise limiting) depending on the relative strengths of input and output noise.

As mentioned in our response to Reviewer 1, we clarified our language throughout the Discussion to emphasize that the efficient coding regimes are a general feature of information optimization, and neither regime is a priori restricted to peripheral vs central processing. Thus, even in sensory periphery, there might exist special regimes when the signal is input-noise dominated. The best known example is perhaps that of night vision, when photon shot noise is no longer negligible and qualitatively reshapes peripheral information processing. Instead of decorrelation by center-surround receptive fields, receptive fields averaging the signal, i.e., they apply gain that enhances the correlations already present in the input in order to fight the detrimental effects of input noise. These special considerations, which are relevant for night vision, do not apply generally to the extensively-studied area of daylight vision.

We would like to emphasize clearly that the constraints considered here (nonlinear image features subject to sampling noise) thus differ from those considered by some of the most successful applications of efficient coding in the periphery. Because of these constraints, our finding that sensitivity increases with signal variability differs qualitatively from the findings of these studies, where whitening is optimal (or near optimal, see Doi and Lewicki (2014)) and sensitivity decreases with signal variability. Moreover, since feature extraction is thought to be one of the main tasks of central vision, the sampling noise constraint is likely to represent a general aspect of cortical processing, rather than a special case as for night vision in the periphery.

In addition to changes in the text, we have added a fourth figure that illustrates these two regimes in a simple parallel-channel model. As an illustration of the regime in which output noise dominates, we discuss the well-known suppression of low spatial frequencies (Discussion). We then hypothesize that the other regime (input noise dominating) is relevant for central processing (Discussion). We discuss the specific application of efficient coding in this context, which differs from an application in the sensory periphery. Here, we apply efficient coding to complex nonlinear features (counts of glider colorings), where sampling fluctuations in glider counts provide a source of input noise.

We removed language that dichotomizes peripheral vs central processing, such as “it applies in a different guise and makes different predictions” (formerly the final sentence of the Abstract). Instead, we call upon the previous successes of efficient coding in the periphery in order to highlight one particular regime of efficient coding (whitening), and we use these examples to contrast the qualitative findings of our study.

What was the exact task being performed by the subjects?

We added the following clarification about the psychophysical task, and we refer the reader to Methods, Psychophysical methods, for additional details:

Results: “To characterize perceptual sensitivity to different statistics, we isolated them in synthetic visual images and used a figure/ground segmentation task (Figure 2). We used a four-alternative forced-choice task in which stimuli consisted of a textured target and a binary noise background (or vice-versa). Each stimulus was presented for 120ms and was followed by a noise mask. Subjects were then asked to identify the spatial location (top, bottom, left, or right) of the target. Experiments were carried out for synthetic stimuli in which the target or back- ground was defined by varying image statistic coordinates independently (Figure 2 shows examples of gamuts from which stimuli are built).”

*What would*
Figure 1
*look like for IId samples from white (or colored) noise?*

When natural images are replaced with white noise, the variation is identical across individual image statistics. (The same is true for colored noise, since our processing pipeline includes whitening.) While this is mathematically guaranteed, it may not be obvious, and we agree that showing it is helpful. We now include a sentence within the main manuscript:

Results: “Interestingly, third-order correlations are the least variable across image patches. An analogous analysis performed on white noise yields a flat distribution with considerably smaller standard deviation values (See Methods, Analysis variants for Penn Natural Image Database, and Figure 1—figure supplement 3 for comparison). These (and subsequent) findings are preserved across different choices of image analysis parameters...”

We have also included an additional Figure in Methods, Analysis variants for Penn Natural Image Database that illustrates this comparison, along with a corresponding addition to the text:

Methods “In Figure 1—figure supplement 3, we show that the relative variation in different image statistics (first shown in Figure 1) is not an artifact of our image analysis pipeline, as the pattern of variation is destroyed if white-noise image patches are instead used.”

Figure 1—figure supplement 3. Image statistics along single coordinate axes for white-noise patches. The robustly observed statistical structure of natural scenes (open circles) is completely absent from the same analysis performed on samples of white noise (shaded circles). The inset shows that this holds across analysis parameters.

How do the results change for images pixelated along a grid that is 45 degrees tilted relative to the vertical and horizontal directions?

Of course we predict that there would still be a close correspondence between image statistics and psychophysics. But testing this; and doing so in a way that accurately isolates what would likely be a subtle effect of the grid rotation, is unfortunately not practical. Re-photographing the images with an oblique sensor would be required to avoid the artifacts that would arise from digital rotation and resampling. Collecting a parallel set of psychophysical data would require approximately 200,000 additional psychophysical judgments (the better part of a year at a humane pace). We hope the reviewer understands.

*Related to this, could the√2 differences in the degree of variation for the vertical/horizontal and diagonal beta data points in*
Figure 3
*be due to the difference in the distance between the points in question given the orientation of the pixelation grid?*

It is an interesting observation, but there is more going on than just a difference in center points: for vertical/horizontal two-point sensitivities, the checks involved share a common edge, while for the diagonal case, they only share a corner. Conversely, other experiments show that increasing the spacing between

checks (in either the cardinal or diagonal directions) has very little effect on the sensitivities, over a fivefold range including the range used here (Conte et al., 2014). So we think that the finding that the √ 2 is likely to be a coincidence. We’d therefore prefer not to mention this point, ratio is approximately as mentioning it but then describing the above evidence would likely be viewed as a distraction.

But we do now emphasize that the diagonal and cardinal directions differ by configuration and not just orientation, so that it is clear that the difference in sensitivities is not simply an “oblique effect.” We have clarified this with the following addition to the main text:

Results: “Note that the difference between the sensitivities in the horizontal and vertical directions (β− and β|) vs the diagonal directions (β\ and β/) is not simply an “oblique effect”, i.e., a greater sensitivity cardinally- vs obliquely-oriented contours (Campbell et al., 1966). Horizontal and vertical pairwise correlations differ from the diagonal pairwise correlations in more than just orientation: checks involved in horizontal and vertical pairwise correlations share an edge, while checks involved in diagonal pairwise correlations only share a corner. Correspondingly, the difference in sensitivities for horizontal and vertical correlations vs diagonal correlations is approximately 50%, which is much larger than the size of the classical oblique effect (10–20%, see Campbell et al. (1966)).”